Descriptions of four new species of Minyomerus Horn, 1876 sec. Jansen & Franz, 2018 (Coleoptera: Curculionidae), with notes on their distribution and phylogeny

Jansen M. Andrew 1 majanse1@asu.edu
http://orcid.org/0000-0001-7089-7018 Franz Nico M. 2
1 School of Life Sciences, Arizona State University , Tempe, AZ , USA
2 ASU Natural History Collections, Arizona State University , Tempe, AZ , USA
Morrone Juan J.
Electronic publication date: 2018 Oct 16
Publication date: 2018
Volume: 6
Electronic Location ID: e5633
Received 2018 May 19; Accepted 2018 Aug 17
Copyright: © 2018 Jansen & Franz
Copyright year: 2018
Copyright holder: Jansen & Franz
License: This is an open access article distributed under the terms of the Creative Commons Attribution License, which permits unrestricted use, distribution, reproduction and adaptation in any medium and for any purpose provided that it is properly attributed. For attribution, the original author(s), title, publication source (PeerJ) and either DOI or URL of the article must be cited.
License URL: https://creativecommons.org/licenses/by/4.0/

Keywords: Biogeography, Desert, Entiminae, Region connection calculus, New species, Weevils, Cladistics, Concept taxonomy, Minyomerus, Curculionidae

Funding: The National Science Foundation DEB-1155984 The United States Department of Agriculture—Agricultural Research Service Agreement 58-1275-1-335 This research was funded in part by the National Science Foundation (DEB-1155984) and the United States Department of Agriculture—Agricultural Research Service (Agreement 58-1275-1-335). There was no additional external funding received for this study. The funders had no role in study design, data collection and analysis, decision to publish, or preparation of the manuscript.

==============================
This contribution adopts the taxonomic concept approach, including the use of taxonomic concept labels (name sec. [according to] source) and region connection calculus-5 (RCC–5) articulations and alignments. Prior to this study, the broad-nosed weevil genus Minyomerus Horn, 1876 sec. Jansen & Franz, 2015 (Curculionidae [non-focal]: Entiminae [non-focal]: Tanymecini [non-focal]) contained 17 species distributed throughout the desert and plains regions of North America. In this review of Minyomerus sec. Jansen & Franz, 2018, we describe the following four species as new to science: Minyomerus ampullaceus sec. Jansen & Franz, 2018 (henceforth: [JF2018]), new species, Minyomerus franko [JF2018], new species, Minyomerus sculptilis [JF2018], new species, and Minyomerus tylotos [JF2018], new species. The four new species are added to, and integrated with, the preceding revision, and an updated key and phylogeny of Minyomerus [JF2018] are presented. A cladistic analysis using 52 morphological characters of 26 terminal taxa (5/21 outgroup/ingroup) yielded a single most-parsimonious cladogram (Length = 99 steps, consistency index = 60, retention index = 80). The analysis reaffirms the monophyly of Minyomerus [JF2018] with eight unreversed synapomorphies. The species-group placements, possible biogeographic origins, and natural history of the new species are discussed in detail.

Introduction

This phylogenetic study follows Jansen & Franz (2015) in the use of the taxonomic concept approach; see Franz & Peet (2009), Franz et al. (2016a, 2016b). Accordingly:

Taxonomic concept labels—i.e., the taxonomic name sec. (according to) author or source (year)—are used whenever we identify one specific usage of the taxonomic name. Examples: Minyomerus Horn, 1876 sec. Jansen & Franz, 2015 (henceforth: [JF2015]) and Minyomerus Horn, 1876 sec. Jansen & Franz, 2018 (henceforth: [JF2018]). We also employ this convention to express nomenclatural relationships.

Solely the taxonomic name—without the sec. annotation—is used to refer to the cumulative history (origin to present) of taxonomic concept labels in which that name participates. Example: Minyomerus Horn, 1876.

The annotation [non-focal] is added to taxonomic names whose meanings are not under scrutiny in the present context; such as names for higher-level weevil groups and associated plants (exempting common names). Example: Tanymecini Lacordaire, 1863 [non-focal].

The weevil genus Minyomerus Horn, 1876 [JF2018] remains currently assigned to the tribe Tanymecini Lacordaire, 1863 [non-focal], subtribe Tanymecina Lacordaire, 1863 [non-focal] (Curculionidae [non-focal]: Entiminae [non-focal]—higher-level classification in accordance with Alonso-Zarazaga & Lyal, 1999 and Bouchard et al., 2011). A recent phylogenetic revision of the genus Minyomerus [JF2015] recognized a total of 17 described species, distributed throughout the desert and plains regions of North America (Jansen & Franz, 2015).

Members of the genus Minyomerus [JF2018] are phytophagous, and may be found on a variety of host plants, especially the creosote bush Larrea tridentata (DC) Coville [non-focal] (Zygophyllaceae [non-focal]), broomweed Gutierrezia Lagasca [non-focal] (Asteraceae [non-focal]), sagebrush Artemisia Linnaeus [non-focal] (Asteraceae [non-focal]), and occasionally on other various members of Asteraceae [non-focal] (Jansen & Franz, 2015). While many species appear to be generalists, the adults are consistently observed on the leaves and branches of the host, feeding on the leaf tissue. All other life stages remain unknown. Species of Minyomerus [2018] are commonly found in deserts throughout western North America; including the Mojave, Sonoran, Chihuahuan, and Great Basin Deserts. However, their distributional range extends throughout the semi-arid regions of the Great Plains, the Colorado Plateau, and Baja California, México (O’Brien & Wibmer, 1982; Jansen & Franz, 2015). The adults are flightless, as the hind wings and associated flight structures of all species are either greatly reduced or not readily apparent in dissection.

Minyomerus [JF2018] belongs to the broad-nosed weevils, subfamily Entiminae [non-focal], on the basis of having a short, broad rostrum and dehiscent mandibular process (Marvaldi, 1997; Anderson, 2002; Oberprieler, Marvaldi & Anderson, 2007; Oberprieler, Anderson & Marvaldi, 2014; Marvaldi et al., 2014). The adults are clothed in appressed, circular scales, generally in earth-tones from white to dark brown, with sub-recumbent to erect, interspersed setiform scales (“setae”) arranged in rows on the elytral intervals. Their body length can range from 2.8 to 6.0 mm (Jansen & Franz, 2015). The genus has been classified in the tribe Tanymecini [non-focal] based on the presence of post-ocular vibrissae that project anteriorly from the anterior prothoracic margin, although the exact placement and sister taxa of this genus within the tribe are currently unknown (Howden, 1959, 1970, 1982; Jansen & Franz, 2015).

Minyomerus [JF2015] was circumscribed by a unique combination of synapomorphic traits, described by Jansen & Franz (2015) as follows: The integument is covered by appressed scales that are sub-circular and overlap posteriorly.

The nasal plate is present as a broad, scale-covered, chevron-shaped ridge demarcating the epistoma.

A sulcus posteriad of nasal plate is present.

The scrobe is sub-equal in length to the funicle and club combined.

The head is directed slightly ventrally.

The metatibial apex lacks setiform bristles yet displays bristles that are shorter to sub-equal in length to the surrounding setae and conical to lamelliform.

The mesotarsi are slightly shorter than the mesotibiae.

All tarsi lack pads of setiform setae but have stout, spiniform setae.

The following additional characters are useful for identifying members of Minyomerus [JF2018], especially when differentiating the former from other genera of Tanymecini [non-focal] that may be found together in the same desert habitats; viz. Isodrusus Sharp, 1911 [non-focal], Isodacrys Sharp, 1911 [non-focal], and Pandeleteinus Champion, 1911 [non-focal] (see also Anderson, 2002): The intercoxal process of the prosternum is medially divided into two halves, with the procoxae apparently contiguous in most.

The elytral humeri are rounded rather than angled and protruding.

The profemora are not dilated and lack spines.

The protibiae are ventrally excavated by a longitudinal groove or concavity.

A distinct scrobe is present and directed ventrad of the eye, with a more or less apparent tooth formed by an overhang of the dorsal margin.

Following the publication of a monographic revision of Minyomerus [JF2015], we have discovered four additional, undescribed species. These are known to us only from limited numbers of specimens, yet are well circumscribed by—i.e., intensionally included in (see Franz & Peet, 2009)—the recent generic delimitation of Minyomerus [JF2015]. In other words, the addition of these new species has not required altering the intensional, property-based definition of the genus-level concept as circumscribed in Jansen & Franz (2015) (see Phylogenetic Results). Our region connection calculus-5 (RCC–5) alignments (see RCC–5 Alignments) reflect this genus-level concept congruence while also showing which classificatory and phylogenetic structures have changed (Figs. 1–3). The precise use of the taxonomic concept labels in accordance with either [JF2015] or [JF2018] is meant to minimize the creation of new taxonomic concept labels (to counter label “inflation”; see Franz & Peet (2009)), while reflecting explicitly which taxonomic concepts we consider as relevantly new and unique to the present study.

Figure 1 Intensional RCC–5 alignment of the rank-only classifications of Minyomerus [JF2018]/[JF2015].

See also Jansen & Franz (2015) and Supplemental Information SI2. Taxonomic concept labels such as Minymerus microps [JF2015] are abbreviated as “2015.Minyomerus_microps.” Relaxation of the coverage constraint is indicated with the prefix “nc_” (no coverage). Congruent concept regions (T2 and T1) are shown as gray rectangles, concepts regions unique to the later taxonomy (T2) are shown as green rectangles, and concept regions unique to the earlier taxonomy (T1) are shown as yellow octagons. Articulations of inverse proper inclusion (<) and overlap (><), where present, are also shown.

Figure 2 Intensional RCC–5 alignment of the phylogenies of Minyomerus [JF2018]/[JF2015]—whole-concept resolution with overlap.

See also Supplemental Information SI3. Seven overlapping articulations are inferred. For further discussion, see the RCC–5 Alignments section.

Figure 3 Intensional RCC–5 alignment of the phylogenies of Minyomerus [JF2018]/[JF2015]—split-concept resolution.

See also Supplemental Information SI4. The seven overlapping articulations of the alignment displayed Fig. 2 are resolved into their constituent split regions. That is, if regions A and B overlap, the three resulting split regions are labeled A\b (“A, not b”), A*B (“A and B”), and B\a (“B, not a”). Five split-concept regions can only be named using this convention, and are salmon-colored in the alignment visualization.

Here, we describe the four newly found species of Minyomerus [JF2018] and provide images of the holotypes and of dissected genitalia for the purpose of identification. We additionally conduct a morphological phylogenetic analysis of the genus to clarify the placement of these new taxa within Minyomerus [JF2018], based on the analysis provided in our previous work. An emended identification key to the species of Minyomerus [JF2018] is given, along with an updated species checklist. Where possible, we make note of host plant records, and briefly discuss the geographic distributions of the herein described species. A more extensive discussion of the habits, distribution, and delimitation of the genus Minyomerus [JF2015] and all of its constituent species is provided in Jansen & Franz (2015).

Materials and Methods

The methods used in this manuscript are generally consistent with Jansen & Franz (2015). Relevant updates are detailed below. In particular, we retain the format for the species descriptions, emphasizing only those characters that vary significantly from the generic circumscription of Minyomerus [JF2015].

Acquisition of museum specimens

The set of specimens used in Jansen & Franz (2015) was supplemented with material from the following collections, using the codens of Arnett, Samuelson & Nishida (1993): CMNC Canadian Museum of Nature Collection, Ottawa, Ontario, Canada

TAMU Texas A&M University, College Station, Texas, USA

USNM National Museum of Natural History, Washington, D.C., USA

Georeferencing of localities was performed with Google Earth (Google Inc., 2018), following the WGS 84 standard, and reported in decimal degrees. Taxonomic names for associated host plants, as noted following each species account, are used in accordance with Munz & Keck (1973) and SEINet (2018).

Morphological analysis

Our systematic and descriptive approach is complementary to Jansen & Franz (2015), which in turn follows Franz (2010a, 2010b, 2012). The terminology for exterior morphology is in general accordance with De La Torre-Bueno, Nichols & Tulloch (1989). Additional morphological terms specific to broad-nosed weevils (Entiminae [non-focal]) were used as follows: Ting (1936) and Morimoto & Kojima (2003) for mouthparts; Thompson (1992) for tibial apices and abdominal segments; and Oberprieler, Anderson & Marvaldi (2014) and Howden (1995) for male and female terminalia.

Measurements were taken with a Leica M205 C stereomicroscope and associated software, Leica Application Suite, version 4.1.0. Overall body length and width were measured in dorsal view as the maximum distance between the rostral and elytral apices, and the maximum width of both elytra, respectively. Rostral length was measured in dorsal view as the distance between the epistomal apex and the anterior margin of the eyes. Rostral width was measured in dorsal view as the maximum distance between the dorsal margins of the rostrum near the point of antennal insertion. Pronotal length was measured in dorsal view as the length along the midline between the anterior and posterior margins. The width of an individual elytron was measured in dorsal view as the maximum distance between the lateral margin and the elytral suture. Other length and width measurements were also performed in dorsal orientation, using the maximum length and width of the corresponding structure (profemur, protibia, elytron, and aedeagus). Images of mouthparts and terminalia were produced with the Leica microscope equipment, while habitus photographs were created with a Visionary Digital Passport II system using a Canon EOS Mark 5D II camera.

The herein newly recognized species of Minyomerus [JF2018] were delimited through application of the phylogenetic species concept sensu Wheeler & Platnick (2000). Species descriptions are in alphabetical order, rather than phylogenetic order, for ease of use. As in Jansen & Franz (2015), the species descriptions represent unique, complementary accounts of the character states observed for each species, including their intra-specific variability, but excepting characters invariant within the genus-level concept of Minyomerus [JF2015]. Likewise, descriptions of males emphasize characters that are variable and sufficiently different from those of the females to merit recognition. The key to identifying species of Minyomerus [JF2018] is arranged with emphasis being placed on the most readily observable diagnostic characters. This manuscript is arranged with the species descriptions appearing first, followed by the key to species, and then by the phylogenetic and RCC–5 alignment results.

Phylogenetic analysis

The morphological cladistic analysis includes 26 terminal taxa; with 21 ingroup and five outgroup terminals. The ingroup terminals were represented by 17 species previously assigned to Minyomerus [JF2015] and four newly recognized species. In keeping with our previous analysis, we sampled outgroups fairly broadly while remaining focused on North American lineages that are putative close relatives of the ingroup (Jansen & Franz, 2015; Nixon & Carpenter, 1993).

Although the tribe Tanymecini [non-focal] is cosmopolitan, the majority of New World species diversity in the tribe may be found in the subtribe Tanymecina [non-focal] (Alonso-Zarazaga & Lyal, 1999). Thus, four of the five outgroup terminals are represented by species belonging to separate genera in the Tanymecina [non-focal]; viz. Isodacrys buchanani Howden, 1961 [non-focal], Isodrusus debilis Sharp, 1911 [non-focal], Pandeleteinus subcancer Howden, 1969 [non-focal], and Pandeleteius cinereus (Horn, 1876) [non-focal]. Because generic relationships in the Tanymecini [non-focal] remain unresolved, we selected a relatively far-removed taxon to root the cladogram that would nevertheless display states applicable to the ingroup for characters under consideration (Rieppel, 2007; Franz, 2014). To this end we used the North American species Sitona californicus (Fahraeus, 1840) [non-focal], of the tribe Sitonini Gistel, 1856 [non-focal].

The character matrix was edited and phylogenetic results viewed using the WinDada and WinClados interfaces of WinClada, respectively (Nixon, 2002). Characters are numbered in accordance with descriptive sequence used in the species accounts. A “–” symbol indicates inapplicable (character, state), whereas a “?” symbol indicates missing information, for example, due to the unavailability of male specimens or insufficient specimens on hand to permit full dissections. Characters 9, 27, 39, 45–47, 49, and 51 were mapped onto the preferred phylogeny using ACCTRAN optimization (see Agnarsson & Miller, 2008), and the remaining characters had an unambiguous optimization. All multi-state characters but one were coded as additive, as explained beneath the description for each character (see Phylogenetic Results), based on their alignment with the preferred phylogeny. Each alternative coding scheme was tested both alone and in unison with the other multi-state characters to assess their impact on the topology of the preferred phylogeny.

The most-parsimonious tree and character state optimizations were inferred under parsimony using NONA (Goloboff, 1999). An unconstrained heuristic search was conducted using the commands: hold 100001, mult*1000, hold/100, with mult*max* selected. Bootstrap support was inferred in WinClada using the parameters of 1000 replications, hold 1000, hold/100, mult*10, Don’t do max*, and Save consensus. Finally, Bremer support values (bsv) (Bremer, 1994) and relative fit difference (rfd) (Goloboff & Farris, 2001) were calculated in NONA using the commands: hold 1001, sub 20, bs for bsv, and bs* for rfd, respectively (Goloboff, Farris & Nixon, 2008).

The motivation for providing bsv and rfd comes from their respective interpretations, based on how the measures are calculated, per Goloboff & Farris (2001). Both of these indices rely on summation of the number of favorable and contradictory characters when comparing a most-parsimonious tree to a suboptimal tree. If the step length of the ith character (I) of n total characters on the most-parsimonious tree (LMPT) is less than its corresponding step length on the suboptimal tree (LSUB), the character is designated as favorable (fi), but if the opposite is true, the character is designated as contradictory (ci), and expressed formally: (1) I={fiLMPT<LSUBciLMPT>LSUB

Where the number of favorable (F) and contradictory (C) characters are defined, respectively, as: (2) F=∑n=0ifi

(3) C=∑n=0ici

Bremer support values and rfd are then calculated simply as: (4) bsv=F−C

(5) rfd=F−CF×100

The bsv for a node thus indicates how many more characters support a node than contradict it, while the rfd indicates what proportion of the favorable characters are represented by the bsv. Whereas the bsv is as large as the number of characters supporting the node, in excess of the contradicting characters, the rfd can only vary from 0 to 100, as a proportion of the number of supporting characters. By providing both measures, one may quickly discriminate, for example, between a node supported by four characters but contradicted by one character (bsv = 3, rfd = 75), and a node supported by 10 characters but contradicted by seven characters (bsv = 3, rfd = 30).

Taxonomic annotations and RCC–5

In accordance with Jansen & Franz (2015), we use the symbol “=” to indicate nomenclatural synonymy (objective/subjective); and the RCC–5 symbols {==, >, <, ><, !} indicate taxonomic concept articulations. The annotations (INT) and (OST) indicate intensional and ostensive readings of articulations, and AND is used to connect multiple simultaneously recognized provenance relationships. Two intensional alignments are produced as part of this review, that is, one that captures the non-congruence of Minyomerus [JF2018] vs. Minyomerus [JF2015] represented as rank-only classifications (Fig. 1), and another that represents these as fully bifurcated phylogenies with newly assigned clade concept labels, shown in whole-concept resolution (Fig. 2) and in split-concept resolution (Fig. 3); see Franz et al. (2018).

A detailed breakdown of our alignment approach and outcomes using an RCC–5 logic reasoner toolkit (Chen et al., 2014) is provided in the Supplemental Information, SI1–SI4. For further information, see also Jansen & Franz (2015), Franz et al. (2016a, 2016b).

Species distribution modeling

We used the modeling program Maxent, Version 3.4, to generate habitat models for the species of Minyomerus [JF2018] (Figs. 4–7) based on documented occurrence records (Phillips, Dudík & Schapire, 2004; Phillips, Anderson & Schapire, 2006; Elith et al., 2011). The default settings were adjusted to Max number background points = 100,000 and Iterations = 10. Cross-validation was used to leverage all available locality data; however, no models could be created for species with two or fewer documented localities. We selected 19 bioclimatic variables and elevation as Environmental Layers in Maxent, obtained from WorldClim (Hijmans et al., 2005). The layers were downloaded by tile (zones 11–13 and 21–23), with a 30 arc-second resolution (projected using WSG 84) to provide adequate coverage of the full distribution of the genus. Layerwise assembly of tiles was done using QGIS, Version 2.18.16 “Las Palmas,” creating composite maps of six tiles each to use in species distribution modeling (Quantum GIS Development Team, 2018).

Figure 4 Summary map of distributions of new species of Minyomerus [JF2018].

Combined occurrence record and Maxent habitat modeling map for four newly-described species of Minyomerus [JF2018], as indicated in the legend.

Figure 5 Distributions of M. ampullaceus [JF2018] and M. tylotos [JF2018].

Combined occurrence record and Maxent habitat modeling map for M. ampullaceus [JF2018] and M. tylotos [JF2018], as indicated in the legend.

Figure 6 Distributions of M. franko [JF2018].

Combined occurrence record and Maxent habitat modeling map for M. franko [JF2018], as indicated in the legend.

Figure 7 Distributions of M. sculptilis [JF2018].

Combined occurrence record and Maxent habitat modeling map for M. sculptilis [JF2018], as indicated in the legend.

The rasterized predictive probabilities were imported into QGIS, where each file was designated a specific color. Each pixel in the raster was assigned a linearly interpolated saturation of that color, with increasing saturation denoting an increased probability of successful prediction of species presence at that point. Pixels with a value below 0.50 were rendered transparent so that the maps only show regions with a greater than 50% chance of successful prediction. The raster files were clipped to remove extraneous predicted regions based on: (1) predictive probability (i.e., removing large areas with only transparent pixels) and (2) geographic extent (accounting for endemicity). For example, a species endemic to the Snake River Valley of Idaho does not require a predictive model for bioclimatically similar habitats in the Chihuahuan Desert. Documented occurrence records are laid over the modeled habitat ranges as colored circles on their respective maps (Figs. 4–7), along with vector layers of country (white) and state (gray) borders (Hijmans et al., 2012).

Nomenclature

The electronic version of this article in portable document format will represent a published work according to the International Commission on Zoological Nomenclature (ICZN), and hence the new names contained in the electronic version are effectively published under that Code from the electronic edition alone. This published work and the nomenclatural acts it contains have been registered in ZooBank, the online registration system for the ICZN. The ZooBank Life Science Identifiers (LSIDs) can be resolved and the associated information viewed through any standard web browser by appending the LSID to the prefix http://zoobank.org/. The LSID for this publication is: urn:lsid:zoobank.org:pub:0AEE5733-06D1-401F-88C9-0D5232FBFC7A. The online version of this work is archived and available from the following digital repositories: PeerJ, PubMed Central, and CLOCKSS.

Minyomerus ampullaceus: Minyomerus franko: Minyomerus sculptilis: Minyomerus tylotos:

Descriptions of New Species

Minyomerus ampullaceus Jansen & Franz sec. Jansen & Franz, 2018; sp. n.

urn:lsid:zoobank.org:act:24943E17-F20E-4E3C-A3A1-A1D4D907B48E

Figures 8–13

Figure 8 Dorsal habitus of M. ampullaceus [JF2018].

Image of female (♀) holotype. Photo credit: Andrew Jansen.

Figure 9 Lateral habitus of M. ampullaceus [JF2018].

Image of female (♀) holotype. Photo credit: Andrew Jansen.

Figure 10 Ventral habitus of M. ampullaceus [JF2018].

Image of female (♀) holotype. Photo credit: Andrew Jansen.

Figure 11 Head and rostrum of M. ampullaceus [JF2018].

Frontal view of female (♀) holotype. Photo credit: Andrew Jansen.

Figure 12 Spermatheca of M. ampullaceus [JF2018].

Genitalia of female (♀) holotype. Photo credit: Andrew Jansen.

Figure 13 Lamina of spiculum ventrale of M. ampullaceus [JF2018].

Sternum VIII of female (♀) holotype. Photo credit: Andrew Jansen.

Diagnosis

Minyomerus ampullaceus [JF2018] is best differentiated from other congenerics by its unique body shape, which most prominently features a strongly constricted, sub-cylindrical pronotum and greatly protuberant elytra; this combination gives the species a distinctly flask- or bottle-shaped appearance. Due to the relatively poor condition of the scales and setae of the holotype, color and setation cannot be reliably used for identification. However, the elytra themselves are unique in shape, and diagnostic, together nearly 2× the width of the pronotum at their widest point, and nearly 3/4× as wide as long in dorsal view. In lateral view the anterior and posterior declivities of the elytra are strongly abrupt, and nearly vertical; most notably, the anterior margin of the elytra projects strongly and characteristically dorsad of its articulation with the posterior pronotal margin. The spermatheca is also quite distinct, having a highly elongate projection of the corpus aligned with midline of the ramus, which is basally tapered and angled at nearly 45° to the corpus.

Description of female

Habitus. Length 3.76 mm, width 1.76 mm, length/width ratio 2.14, widest at anterior 1/3 of elytra. Integument orange-brown to black. Scales with variously interspersed colors ranging from slightly off-white to beige to yellow. Setae recumbent to sub-recumbent, white to brown in color.

Mandibles. Partially covered with white, slightly opalescent scales, with three longer setae, and one shorter seta between these.

Rostrum. Length 0.54 mm, anterior portion 1.5–2× broader than long, rostrum/pronotum length ratio 0.57, rostrum length/width ratio 1.10. Separation of rostrum from head generally obscure. Dorsal outline of rostrum nearly square, anterior half of dorsal surface mesally concave, posterior half coarsely but shallowly punctate to rugose. Rostrum in lateral view nearly square; apical margin broadly bisinuate and emarginate, with two pairs of large vibrissae. Nasal plate defined by Y-shaped, impressed lines, convex, integument partially covered with white scales. Margins of mandibular incision directed ca. 15° outward dorsally in frontal view. Ventrolateral sulci strongly defined, beginning as a narrow sulcus posteriad of insertion point of mandibles, running parallel to scrobe, terminating in a ventral fovea.

Antennae. Small tooth formed by overhanging dorsal margin of scrobe directly ventrad of margin of eye. Scape extending to posterior 1/3 of eye. Funicular segments V–VII and club missing.

Head. Eyes globular, anterodorsal margin of each eye feebly impressed, posterior margin elevated from lateral surface of head; eyes separated in dorsal view by 4× their anterior–posterior length, set off from anterior prothoracic margin by 1/3 of their anterior–posterior length. Head without any transverse post-ocular impression.

Pronotum. Length/width ratio 0.88; widest near midpoint. Anterior margin slightly arcuate, lateral margins curved and widening into a bulge just anteriad of midpoint of pronotum, posterior margin straight, with a slight mesal incurvature. Pronotum in lateral view with setae that reach beyond anterior margin by 1/2 of their length; these setae becoming evenly longer and more erect laterally, reaching a maximum length equal to 1/2 of length of eye. Anterolateral margin with a reduced tuft of 6–7 post-ocular vibrissae present, emerging near ventral 1/2 of eye, and stopping just below ventral margin of eye; vibrissae sub-equal in length at 1/3 of anterior–posterior length of eye, except for three vibrissae achieving a maximum length similar to anterior–posterior length of eye.

Scutellum. Exposed, margins straight.

Pleurites. Metepisternum hidden by elytron.

Thoracic sterna. Mesocoxal cavities separated by 1/4× width of mesocoxal cavity. Metasternum with transverse sulcus not apparent; metacoxal cavities widely separated by ca. 2× their width.

Legs. Profemur/pronotum length ratio 1.04; profemur with distal 1/5 produced ventrally as a rounded projection covering tibial joint; condyle of tibial articulation occupying 4/5 of distal surface and 1/5 length of femur. Protibia/profemur length ratio 0.93; protibial apex with ventral setal comb recessed in an incurved groove; mucro present as a large, black, sub-triangular, medially-projected tooth, which is approximately equilateral and whose sides are sub-equal in length to surrounding setae. Protarsus with tarsomere III 1.25× as long as II; wider than long. Metatibial apex with almond shaped convexity ringed by 10 short, spiniform setae.

Elytra. Length/width ratio 2.66; widest at anterior 1/3; anterior margins jointly almost 2× wider than posterior margin of pronotum and strongly produced dorsally from margin of pronotum; lateral margins evenly rounded until posterior 1/3, more strongly rounded and converging thereafter. Posterior declivity angled at nearly 85° to main body axis. Elytra with 10 complete striae; striae shallow; punctures faint beneath appressed scales, separated by 5–7× their diameter; intervals very slightly elevated.

Abdominal sterna. Ventrite III anteromesally incurved around a fovea located mesally on anterior margin, posterior margin elevated and set off from IV along lateral 1/3 s of its length. Sternum VII mesally 1/2× as long as wide; anterior margin weakly curved.

Tergum. Pygidium (tergum VIII) sub-conical; posterior margin emarginate; medial 1/3 of anterior 3/5 of pygidium less sclerotized.

Sternum VIII. Anterior laminar edges each incurved forming a 115° angle with lateral margin, this angle distinctly sclerotized; posterior 1/2 of lamina porose throughout, laminar arms more sclerotized medially; posterior edge evenly, moderately arcuate.

Ovipositor. Coxites in dorsal view slightly longer than broad, with a medial region that is weakly sclerotized.

Spermatheca. Comma-shaped; collum expanded to form a long, cylindrical projection, sub-equal in length to ramus, 1/3× width of corpus, angled at 45° to corpus, apically with a reduced hood-shaped projection; ramus elongate, bulbous, slightly wider than thickness of corpus, basally constricted to form a short stalk; corpus not greatly swollen; cornu sub-equal in length to corpus and collum, recurved distally to form in inner angle of 60° to corpus, straight and gradually narrowing along basal 2/3, with apical 1/3 abruptly narrowed, angled at 45° to corpus, and tapering to a slight knob.

Description of male

Male not available or known.

Comments. Due to the limited number of specimens of this species, dissections of mouthparts could not be performed.

Etymology. Named in reference to the shape of the body in dorsal view, which appears bottle-shaped due to the large elytra and comparatively cylindrical pronotum—ampullaceus = “flasklike”; Latin adjective (Brown, 1956).

Material examined

Holotype. ♀ “Carlsbad, N.M.; Geococcyx calif; 144640” (USNM).

Distribution. This species is known only from Carlsbad, New Mexico (USA), from an unspecified locality; the location of the city is shown in Fig. 5.

Natural history. No host plant associations have been documented. The label indicates “Geococcyx calif”; this is presumably a reference to Geococcyx californianus (Lesson, 1829) [non-focal] (Cuculidae [non-focal]), the greater roadrunner. We had initially believed that this indicated a specimen found in a roadrunner nest; however, according to our reviewers, the USNM frequently assisted with the identification of insect specimens retrieved from the stomach contents of birds, and thus the specimen was most likely retrieved from the gut contents of a roadrunner. This seems quite likely given the poor external condition of the specimen. It is unknown whether this species is parthenogenetic.

Minyomerus franko Jansen & Franz sec. Jansen & Franz, 2018; sp. n.

urn:lsid:zoobank.org:act:F8C0153E-DF0E-40E0-AF31-EBEA7075D06D

Figures 14–22

Figure 14 Dorsal habitus of M. franko [JF2018].

Image of female (♀) holotype. Photo credit: Andrew Jansen.

Figure 15 Lateral habitus of M. franko [JF2018].

Image of female (♀) holotype. Photo credit: Andrew Jansen.

Figure 16 Ventral habitus of M. franko [JF2018].

Image of female (♀) holotype. Photo credit: Andrew Jansen.

Figure 17 Head and rostrum of M. franko [JF2018].

Frontal view of female (♀) holotype. Photo credit: Andrew Jansen.

Figure 18 Maxilla of M. franko [JF2018].

Dextral maxilla of female (♀) paratype. Photo credit: Andrew Jansen.

Figure 19 Prementum of M. franko [JF2018].

Labium of female (♀) paratype. Photo credit: Andrew Jansen.

Figure 20 Spermatheca of M. franko [JF2018].

Genitalia of female (♀) paratype. Photo credit: Andrew Jansen.

Figure 21 Lamina of spiculum ventrale of M. franko [JF2018].

Sternum VIII of female (♀) paratype. Photo credit: Andrew Jansen.

Figure 22 Aedeagus of M. franko [JF2018].

Genitalia of male (♂) paratype in (A) dorsal view and (B) lateral view. Photo credit: Andrew Jansen.

Diagnosis

Minyomerus franko [JF2018] is readily distinguished from other congenerics by the strikingly long setae of the anterior margin of the pronotum, which project laterally up to 80 from the longitudinal axis of the body and achieve a maximum length at least equaling the diameter of the eye. In addition, the setae lining the dorsal margin of the ocular impression are elongate and reach a length equal to 1/2–3/4× the diameter of the eye. The spermatheca has a short, somewhat bulbous corpus, with the ramus sub-equal in size and perpendicular to the corpus, and the collum is strongly recurved along the basal 1/3 of its length. The aedeagus is relatively short and wide, and is abruptly constricted in the apical 1/5 of its length, thereafter tapered to a rounded point.

Description of female

Habitus. Length 3.10–3.30 mm, width 1.38–1.44 mm, length/width ratio 2.25–2.29, widest at anterior 1/3–1/4 of elytra. Integument orange-brown to black. Scales with variously interspersed colors ranging from slightly off-white or beige to manila/tan to dark coffee brown, in some specimens appearing semi-translucent (in others opaque). Setae linear to slightly apically explanate, appearing minutely spatulate, sub-recumbent to sub-erect, white or brown in color.

Mandibles. Covered with white scales, with three longer setae, and 1–2 shorter setae between these.

Maxillae. Cardo bifurcate at base with an inner angle typically between 90° and 120°, arms of equal length, inner (mesal) arm nearly 1.5× thicker than outer arm, both arms of bifurcation equal in length to apically outcurved arm, glabrous. Stipes sub-quadrate, roughly equal in length to each bifurcation of cardo, with a single lateral seta. Galeo-lacinial complex nearly extending to apex of maxillary palpomere II; complex mesally membranous, laterally sclerotized, with sharp demarcation of sclerotized region separating palpiger from galeo-lacinial complex; setose in membranous area just adjacent to sclerotized region, setae covering 2/3 of dorsal surface area; dorsally with seven apicomesal lacinial teeth; ventrally with four reduced lacinial teeth. Palpiger with a single lateral seta, otherwise glabrous and evenly sclerotized throughout.

Maxillary palps. I apically oblique, apical end forming a 45° angle with base, with two apical setae; II sub-cylindrical, with one apical seta.

Labium. Prementum roughly trapezoidal; apical margins angulate, ventral margin gently sinuate, dorsal margin straight; lateral margins feebly incurved near posterior margin; basal margin arcuate. Labial palps 3-segmented, I with apical 2/3 projecting beyond margin of prementum, exceeding apex of ligula; III slightly longer than II.

Rostrum. Length 0.46–0.48 mm, anterior portion 1.75–2.25× broader than long, rostrum/pronotum length ratio 0.58–0.59, rostrum length/width ratio 1.21–1.26. Separation of rostrum from head generally obscure. Dorsal outline of rostrum sub-rectangular, anterior half of dorsal surface feebly impressed, posterior half coarsely but shallowly punctate to rugose. Rostrum in lateral view nearly square; apical margin bisinuate and emarginate, with two large vibrissae. Nasal plate defined by broad, V-shaped, shallowly impressed lines, anteromesally slightly convex, integument partially covered with white scales. Margins of mandibular incision directed ca. 15° outward dorsally in frontal view. Ventrolateral sulci weakly defined (or entirely absent in some specimens) as a broad concavity dorsad of insertion point of mandibles, running parallel to scrobe, becoming flatter posteriorly and disappearing ventrally. Dorsal surface of rostrum with short, linear, median fovea. Rostrum ventrally lacking sulci at corners of oral cavity.

Antennae. Small tooth formed by overhanging dorsal margin of scrobe anterior to margin of eye by 1/5 of length of eye. Scape nearly extending to posterior 1/4 of eye. Terminal funicular antennomere lacking appressed scales, having instead a covering of apically-directed pubescence with interspersed sub-erect setae. Club nearly 3× as long as wide.

Head. Eyes globular to slightly elongate, slanted ca. 35° antero-ventrally; eyes separated in dorsal view by 4× their anterior–posterior length, set off from anterior prothoracic margin by 1/3 of their anterior–posterior length. Head without any transverse post-ocular impression.

Pronotum. Length/width ratio 0.84–0.86; widest near anterior 1/3, between anterior constriction and midpoint. Anterior margin arcuate, lateral margins curved and widening into a slight bulge just anteriad of midpoint of pronotum, posterior margin straight, with a slight mesal incurvature. Pronotum in lateral view with setae that reach just beyond anterior margin, angled laterally at 45–80° to longitudinal axis, and strikingly long; these setae becoming evenly longer and more angled laterally, reaching a maximum length nearly equal to length of eye. Anterolateral margin with a reduced tuft of five post-ocular vibrissae present, emerging near ventral 1/2 of eye, and stopping just below ventral margin of eye; vibrissae sub-equal in length at 1/3× anterior–posterior length of eye, except for one vibrissa achieving a maximum length similar to anterior–posterior length of eye.

Scutellum. Narrowly exposed, with visible area approximately equal to length of appressed scales, margins straight.

Pleurites. Metepisternum nearly hidden by elytron except for triangular extension.

Thoracic sterna. Mesocoxal cavities separated by 1/3× width of mesocoxal cavity. Metasternum with transverse sulcus not apparent; metacoxal cavities widely separated by ca. 2× their width.

Legs. Profemur/pronotum length ratio 1.01–1.02; profemur with distal 1/5 produced ventrally as a sub-rectangular projection covering tibial joint; condyle of tibial articulation occupying 4/5 of distal surface and 1/5 length of femur. Protibia/profemur length ratio 0.86–0.89; protibial apex with ventral setal comb recessed in a subtly incurved groove; mucro present as a large, black, sub-triangular, medially-projected tooth, which is approximately equilateral and whose sides are sub-equal in length to surrounding setae. Protarsus with tarsomere III 2× as long as II; wider than long. Metatibial apex with almond shaped convexity ringed by 8–9 short, spiniform setae.

Elytra. Length/width ratio 3.08–3.20; widest at anterior 1/3–1/4; anterior margins jointly 1.5× wider than posterior margin of pronotum; lateral margins sub-parallel to slightly rounded after anterior 1/3, more strongly rounded and converging in posterior 1/3. Posterior declivity angled at 70–85° to main body axis. Elytra with 10 complete striae; striae shallow; punctures faint beneath appressed scales, separated by 5–7× their diameter; intervals very slightly elevated.

Abdominal sterna. Ventrite III anteromesally incurved around a fovea located mesally on anterior margin, posterior margin elevated and set off from IV along lateral 1/3 s of its length. Sternum VII mesally 1/2× as long as wide; setae darkening, lengthening, and becoming more erect in posterior 2/3; anterior margin weakly curved.

Tergum. Pygidium (tergum VIII) sub-cylindrical; medial 1/3 of anterior 2/3 of pygidium less sclerotized.

Sternum VIII. Anterior laminar edges each incurved forming a 140° angle with lateral margin; slightly less sclerotized medially between arms of bifurcation; posterior edge subtly incurved medially.

Ovipositor. Coxites 1.5× as long as broad, glabrous; styli 1/2× as long as coxites. Genital chamber apically sclerotized.

Spermatheca. Comma-shaped; collum short, apically with a large, hood-shaped projection angled at ca. 60° to ramus, nearly equal in length and continuously aligned with curvature of bulb of ramus; collum sub-contiguous with, and angled at 90° to ramus; ramus elongate, sub-cylindrical to slightly bulbous, 4/5× thickness of corpus; corpus swollen, 1.25× thickness of ramus and 1.5× thickness of cornu; cornu elongate, strongly recurved in basal 1/3, nearly straight thereafter and narrowing apically, abruptly narrowed in apical 1/3 with apex angled at 30° to corpus.

Description of male

Similar to female, except where noted.

Habitus. Length 2.47–2.81 mm, width 0.99–1.24 mm, length/width ratio 2.27–2.49. Rostrum length 0.30–0.42 mm, rostrum/pronotum length ratio 0.44–0.53, rostrum length/width ratio 1.00–1.08. Pronotum length/width ratio 0.91–1.00. Profemur/pronotum length ratio 0.87–0.90, protibia/profemur length ratio 0.87–0.97. Elytra length/width ratio 3.00–3.10.

Elytra. Elytral declivity more angulate than female on average, forming an 80° angle to main body axis, but otherwise as in female.

Abdominal sterna. Sternum VII 2/5-1/2× as long as wide, posterior margin arcuate mesally.

Tergum. Pygidium (tergum VIII) with posterior 1/3 punctate; anterior 2/3 rugose.

Sternum IX. Spiculum gastrale 2× length of aedeagal pedon. Laminar alae located on lateral 1/4 of posterior margin.

Aedeagus. Length/width ratio 2.78–3.16; lateral margins very slightly converging posteriorly, abruptly constricted and more strongly converging in apical 1/5. Pedon in lateral view becoming gradually narrower posteriorly in anterior 1/2, ventral margins in posterior 1/2 abruptly curving to meet dorsal margins at a rounded apical point. Flagellum with large, elonage, tortuous apical sclerite, sclerite nearly as long as pedon, with complex, asymmetrical interior structure.

Etymology. Named in reference to the long, somewhat unkempt, erect setae on the anterior margin of the pronotum—franko = “free”; Old High-German adjective (Brown, 1956).

Material examined

Holotype. ♀ “MEX: S.L.P 1 km N.; Entronque El Huizache; 1,493 m 2.VI.87; R. Anderson, Sphaeralcea; hastula A. Gray” [non-focal] (CMNC).

Paratypes. Same label information as female holotype (CMNC: 1 ♀, 1♂; TAMU: 2 ♂); “MEXICO: S.L.P; 19.6 mi. n. Huizache; July 25, 1976; Peigler, Gruetzmacher,; R&M Murray, Schaffner” (CMNC: 1 ♂); “MEXICO: San Luis Potosi; Entronque el Hulzache; June 2, 1987; R. Turnbow” (USNM: 1♀; CMNC: 1 ♂); “MEXICO: Tamaulipas; 8.8 mi. ne. Jaumave; October 10, 1973; Gaumer & Clark” (TAMU: 2♀); “9 mi east Santo; Domingo, S.L.P.,; Mexico XI-14-68; Veryl V. Board” (TAMU: 2 ♂).

Distribution. This species has been found in San Luis Potosí and Tamaulipas (Mexico). It is likely to be found throughout the Chihuahuan Desert and arid regions of south-central Mexico based on habitat similarity (Fig. 6).

Natural history. Associated with spear globemallow Sphaeralcea hastulata A. Gray [non-focal] (Malvaceae [non-focal]). The indication of “Sphaeralcea hastula A. Gray” is not a valid name and appears to be a misspelling of Sphaeralcea hastulata [non-focal].

Minyomerus sculptilis Jansen & Franz sec. Jansen & Franz, 2018; sp. n.

urn:lsid:zoobank.org:act:EA0B1AD9-68F2-4409-A0F8-903B0DA0FFF9

Figures 23–29

Figure 23 Dorsal habitus of M. sculptilis [JF2018].

Image of female (♀) holotype. Photo credit: Andrew Jansen.

Figure 24 Lateral habitus of M. sculptilis [JF2018].

Image of female (♀) holotype. Photo credit: Andrew Jansen.

Figure 25 Ventral habitus of M. sculptilis [JF2018].

Image of female (♀) holotype. Photo credit: Andrew Jansen.

Figure 26 Head and rostrum of M. sculptilis [JF2018].

Frontal view of female (♀) holotype. Photo credit: Andrew Jansen.

Figure 27 Spermatheca of M. sculptilis [JF2018].

Genitalia of female (♀) paratype. Photo credit: Andrew Jansen.

Figure 28 Lamina of spiculum ventrale of M. sculptilis [JF2018].

Sternum VIII of female (♀) paratype. Photo credit: Andrew Jansen.

Figure 29 Aedeagus of M. sculptilis [JF2018].

Genitalia of male (♂) paratype in (A) dorsal view and (B) lateral view. Photo credit: Andrew Jansen.

Diagnosis

Minyomerus sculptilis [JF2018] is best distinguished from other congenerics, especially Minyomerus imberbus Jansen & Franz, 2015 [JF2015], by a combination of characters, as follows. The interspersed setae on the body are linear and either brown or white. The anterior margin of the pronotum bears a reduced tuft of post-ocular vibrissae. The head is barely elevated between the eyes. The ventrolateral sulci of the rostrum are well defined. The lateral face of each elytron has the intervals raised and well sculpted in appearance. The spermatheca is distinct and has an elongate, annulate, basally tapered ramus, which is slightly thinner than corpus. The cornu is strongly recurved in the basal half, giving it a uniquely sinuate appearance. Both the corpus and cornu terminate in large, hood-shaped, explanate projections equal in size to the ramus. The aedeagus is elongate, acutely angulate, and narrowing toward the apex more strongly in the region of the ostium.

Description of female

Habitus. Length 3.39–3.70 mm, width 1.33–1.58 mm, length/width ratio 2.34–2.55, widest at anterior 1/5 of elytra. Integument orange-brown to black. Scales with variously interspersed colors ranging from slightly off-white or beige to golden brown to dark coffee brown. Setae sub-recumbent to sub-erect, white to brown in color.

Mandibles. Covered with white scales, with three longer setae, and one shorter seta between these.

Rostrum. Length 0.50–0.59 mm, anterior portion ca. 1.5× broader than long, rostrum/pronotum length ratio 0.66–0.67, rostrum length/width ratio 1.43–1.48. Separation of rostrum from head generally obscure. Dorsal outline of rostrum nearly square, anterior half of dorsal surface mesally concave, posterior half coarsely but shallowly punctate to rugose. Rostrum in lateral view nearly square; apical margin bisinuate and emarginate, with two pairs of large vibrissae. Nasal plate defined by Y-shaped, impressed lines, convex, integument covered with white scales. Margins of mandibular incision directed ca. 15–20° outward dorsally in frontal view. Ventrolateral sulci strongly defined, beginning as a narrow sulcus posteriad of insertion point of mandibles, running parallel to scrobe, terminating in a ventral fovea.

Antennae. Dorsal margin of scrobe overhanging broadly (not forming a minute tooth). Funicle slightly longer than scape. Scape extending to posterior 1/4 of eye. Club nearly 3× as long as wide.

Head. Eyes globular, anterodorsal margin of each eye impressed, posterior margin slightly elevated from lateral surface of head; eyes separated in dorsal view by 5× their anterior–posterior length, set off from anterior prothoracic margin by 1/4 of their anterior–posterior length. Head between eyes rugose and slightly bulging.

Pronotum. Length/width ratio 0.85–0.87; widest near anterior 2/5. Anterior margin arcuate, subtly incurved mesally, and somewhat produced dorsally; anterior constriction broad, posterior margin slightly arcuate. Pronotum in lateral view with setae that reach beyond anterior margin; these setae becoming slightly longer and more erect laterally. Anterolateral margin with a reduced tuft of 3–6 post-ocular vibrissae present, emerging near ventral 1/2 of eye, and stopping just below ventral margin of eye; vibrissae varying in length from 1/2× anterior–posterior length of eye to a maximum length similar to anterior–posterior length of eye.

Scutellum. Exposed, margins straight.

Pleurites. Metepisternum nearly hidden by elytron except for triangular extension.

Thoracic sterna. Mesocoxal cavities separated by 1/3× width of mesocoxal cavity. Metasternum with transverse sulcus not apparent; metacoxal cavities widely separated by ca. 2× their width.

Legs. Profemur/pronotum length ratio 0.92–1.03; profemur with distal 1/6 produced ventrally as a slightly rounded, sub-rectangular projection covering tibial joint; condyle of tibial articulation occupying 4/5 of distal surface and 1/6 length of femur. Protibia/profemur length ratio 0.87–0.93; protibial apex with ventral setal comb recessed in a subtly incurved groove; mucro not apparent. Protarsus with tarsomere III 1.5× as long as II; wider than long. Metatibial apex with almond shaped convexity ringed by 10–12 short, spiniform setae.

Elytra. Length/width ratio 3.12–3.16; widest at anterior 1/5; anterior margins jointly 1.5–2× wider than posterior margin of pronotum; lateral margins gently converging after anterior 1/5, more strongly converging in posterior 1/4. Posterior declivity angled at 65–70° to main body axis. Elytra with 10 complete striae; striae broadly sculpted; punctures faint beneath appressed scales, separated by 5–7× their diameter; intervals elevated, with every second interval, beginning at elytral suture, more strongly raised than adjacent intervals.

Abdominal sterna. Ventrite III anteromesally incurved around a fovea located mesally on anterior margin, posterior margin elevated and set off from IV along lateral 1/3 s of its length. Sternum VII mesally 2/3× as long as wide; anterior margin straight.

Tergum. Pygidium sub-cylindrical; medial 1/2 of anterior 3/5 of pygidium less sclerotized.

Sternum VIII. Anterior laminar edges of spiculum ventrale each incurved forming a 125° angle with lateral margin; lamina more sclerotized medially; posterior margin medially incurved.

Ovipositor. Coxites as long as broad; styli as long as coxites, glabrous.

Spermatheca. S-shaped; collum short, apically with a large, hood-shaped projection roughly aligned with central axis of corpus, nearly equal in length to bulb of ramus; collum sub-contiguous with, and angled at 30° to ramus; ramus elongate, sub-cylindrical to slightly bulbous, 3/4× thickness of corpus, with a short stalk oriented at ca. 45° to the corpus; corpus swollen, 1.3× thickness of ramus; cornu short, 2.5–3× length or ramus, recurved and strongly arched in basal 1/2, forming an inner angle of ca. 80°, feebly sinuate thereafter, with apical 1/2 expanded, then abruptly constricted near apical 1/4 to a fine point.

Description of male

Similar to female, except where noted.

Habitus. Length 3.10 mm, width 1.22 mm, length/width ratio 2.54. Rostrum length 0.53 mm, rostrum/pronotum length ratio 0.65, rostrum length/width ratio 1.66. Pronotum length/width ratio 0.99. Profemur/pronotum length ratio 1.01, protibia/profemur length ratio 0.82. Elytra length/width ratio 3.18.

Elytra. Elytral declivity slightly less angulate than female, forming a 60° angle to main body axis, but otherwise as in female.

Abdominal sterna. Sternum VII 1/2× as long as wide, posterior margin feebly arcuate mesally.

Tergum. Pygidium (tergum VIII) with mesal 1/3 of posterior margin subtly incurved; posterior 2/3 punctate; anterior 1/3 rugose.

Sternum VIII. Consisting of two sub-triangular sclerites; antero-laterally with a sharply-pointed projection as long as anterior–posterior length of triangular portion of sclerite.

Aedeagus. Length/width ratio 7.00; lateral margins parallel, more strongly converging in region of ostium. In lateral view, width of pedon even throughout in anterior 2/3, ventral margins in posterior 1/3 becoming straight toward apex, then curving to meet dorsal margins at a sharp apical point; apex acutely angulate. Flagellum without apparent sclerite.

Comments. Due to the limited number of specimens of this species, dissections of mouthparts could not be performed.

Etymology. Named in reference to the elevated elytral intervals, which give this species a sculpted appearance—sculptilis = “sculpted”; Latin adjective (Brown, 1956).

Material examined

Holotype. ♀ “Burley, Idaho; #7, 5-20-32; A.[rtemisia] tridentata [non-focal]; David E. Fox” (USNM).

Paratypes. “Milner, Idaho; #5a, 7-9-31; S.[alsola] pestifer; David E. Fox” (CMNC: 1 ♀); “Hazelton, Ida; #10 4/29/30; N.[orta] altissma” (USNM: 1 ♂).

Distribution. This species has been found in three localities along the Snake River in Idaho (USA), and is thought to be endemic to the Snake River Plain (Fig. 7).

Natural history. Associated with big sagebrush Artemisia tridentata Nutt. [non-focal] (Asteraceae [non-focal]), tumbleweed Salsola tragus L. [non-focal] (= Salsola pestifer A. Nelson [non-focal]) (Amaranthaceae [non-focal]), and tall tumblemustard Sisymbrium altissimum L. [non-focal] (= Norta altissima (L.) Britt. [non-focal]) (Brassicaceae [non-focal]).

Minyomerus tylotos Jansen & Franz sec. Jansen & Franz, 2018; sp. n.

urn:lsid:zoobank.org:act:10CD3562-5969-4BCF-ACFE-BB0E5E2BF9A6

Figures 30–36

Figure 30 Dorsal habitus of M. tylotos [JF2018].

Image of female (♀) holotype. Photo credit: Andrew Jansen.

Figure 31 Lateral habitus of M. tylotos [JF2018].

Image of female (♀) holotype. Photo credit: Andrew Jansen.

Figure 32 Ventral habitus of M. tylotos [JF2018].

Image of female (♀) holotype. Photo credit: Andrew Jansen.

Figure 33 Head and rostrum of M. tylotos [JF2018].

Frontal view of female (♀) holotype. Photo credit: Andrew Jansen.

Figure 34 Prementum of M. tylotos [JF2018].

Labium of female (♀) paratype. Photo credit: Andrew Jansen.

Figure 35 Spermatheca of M. tylotos [JF2018].

Genitalia of female (♀) paratype. Photo credit: Andrew Jansen.

Figure 36 Lamina of spiculum ventrale of M. tylotos [JF2018].

Sternum VIII of female (♀) paratype. Photo credit: Andrew Jansen.

Diagnosis

Minyomerus tylotos [JF2018] is most readily distinguished from other congenerics by a combination of characters, as follows. The nasal plate lacks distinct impressions, having instead a poorly defined anteromesal convexity completely and evenly covered with white scales. The frons is protuberant and moderately punctate. The entire body, including the legs, head, and venter, are clothed with brown, linear to minutely apically expanded setae, which are of similar length throughout and appear distinctly undifferentiated and uniform across body regions. The body is somewhat bulky, with the pronotum protuberant laterally and globular in dorsal view. The setae lining the anterodorsal margin of the pronotum uniquely apically explanate, with a longitudinal, medial, ridge-like portion that tapers to either side apicolaterally (visible at high magnification). The lateral margins of the elytra are protuberant anteriorly and sub-parallel along the between anterior 1/5 and posterior 1/3 of their length. The spermatheca has the corpus narrow throughout, equal in thickness to the collum. The ramus is basally stalked and apically bulbous. The collum exhibits a double-bend, and is recurved.

Description of female

Habitus. Length 3.46–3.62 mm, width 1.42–1.54 mm, length/width ratio 2.35–2.44, widest at anterior 1/6 of elytra. Integument orange-brown to black. Scales with variously interspersed colors ranging from slightly off-white or beige to manila/tan to dark coffee brown, in some specimens appearing semi-translucent (in others opaque). Setae linear to apically explanate, appearing minutely spatulate, sub-recumbent to sub-erect, tan to brown in color.

Mandibles. Covered with white scales, with 2–3 longer setae, and 1–3 shorter setae between these.

Maxillae. Cardo bifurcate at base with an inner angle of ca. 90°, arms roughly equal in length and width, arms of bifurcation equal in length to apically outcurved arm. Stipes sub-rectangular, 1.5× wider than long, roughly equal in width to inner arm of bifurcation of cardo, glabrous. Galeo-lacinial complex nearly extending to apex of maxillary palpomere I; complex mesally membranous, laterally sclerotized, with sharp demarcation of sclerotized region separating palpiger from galeo-lacinial complex; setose in membranous area just adjacent to sclerotized region, setae covering 1/2 of dorsal surface area; dorsally with five apicomesal lacinial teeth; ventrally with three reduced lacinial teeth. Palpiger with a single lateral seta, otherwise glabrous, anterior 1/2 membranous, posteriorly sclerotized.

Maxillary palps. I apically oblique, apical end forming a 45° angle with base, with two apical setae; II sub-cylindrical, with one apical seta.

Labium. Prementum roughly pentagonal; apical margins arcuate, medially angulate; lateral margins feebly incurved; basal margin arcuate. Labial palps 3-segmented, I with apical 1/2 projecting beyond margin of prementum, reaching apex of ligula; III slightly longer than II.

Rostrum. Length 0.49–0.50 mm, anterior portion 2.25–2.5× broader than long, rostrum/pronotum length ratio 0.58–0.62, rostrum length/width ratio 1.26–1.32. Separation of rostrum from head generally obscure. Dorsal outline of rostrum nearly square, anterior half of dorsal surface feebly impressed, posterior half coarsely but shallowly punctate to rugose. Rostrum in lateral view nearly square; apical margin strongly bisinuate and emarginate, appearing medially notched, with two large vibrissae. Nasal plate lacking distinct impressions, having instead a poorly defined anteromesal convexity, integument completely and evenly covered with white scales. Margins of mandibular incision directed ca. 25–30° outward dorsally in frontal view. Ventrolateral sulci weakly defined as a broad concavity dorsad of insertion point of mandibles, running parallel to scrobe, becoming flatter posteriorly and disappearing ventrally. Dorsal surface of rostrum with median fovea short and linear, or punctate. Rostrum ventrally with sub-parallel sulci beginning at corners of oral cavity and continuing halfway to back of head.

Antennae. Minute tooth formed by overhanging dorsal margin of scrobe anterior to margin of eye by 1/3 of length of eye. Scape extending to posterior margin of eye. Terminal funicular antennomere lacking appressed scales, having instead a covering of apically-directed pubescence with interspersed sub-erect setae. Club nearly 3× as long as wide.

Head. Eyes globular and somewhat elongate, strongly impressed, slanted ca. 45° antero-ventrally; eyes separated in dorsal view by 4× their anterior–posterior length, set off from anterior prothoracic margin by 1/4 of their anterior–posterior length. Head between eyes punctate and protuberant.

Pronotum. Length/width ratio 0.88–0.89; widest near anterior 2/5; somewhat globular. Anterior margin arcuate, but feebly incurved mesally, lateral margins evenly curved and widening into a bulge just anteriad of midpoint of pronotum, posterior margin straight, with a slight mesal incurvature. Pronotum in lateral view with transverse ventrolateral sulci strongly excavated and distinctly sculptured; with short, recumbent to sub-erect setae that barely attain or reach just beyond anterior margin; these setae becoming shorter and more erect laterally, reaching a maximum length nearly equal to length of eye; dorsally, these setae become uniquely apically explanate, with a longitudinal, medial, ridge-like portion that tapers to either side apicolaterally. Anterolateral margin with a single ocular vibrissa present, emerging near ventral margin of eye; vibrissa achieving a maximum length of 2/5 of anterior–posterior length of eye.

Scutellum. Not exposed.

Pleurites. Metepisternum nearly hidden by elytron except for triangular extension.

Thoracic sterna. Mesocoxal cavities separated by 1/3× width of mesocoxal cavity. Metasternum with transverse sulcus not apparent; metacoxal cavities widely separated by ca. 3× their width.

Legs. Profemur/pronotum length ratio 0.90–0.96; profemur with distal 1/5 produced ventrally as a sub-rectangular projection covering tibial joint; condyle of tibial articulation occupying 4/5 of distal surface and 1/5 length of femur. Protibia/profemur length ratio 0.86–0.91; protibial apex with ventral setal comb recessed in a subtly incurved groove; mucro present as an acute, medially-projected tooth, which is approximately equal in length to surrounding setae. Protarsus with tarsomere III 2× as long as II; wider than long. Metatibial apex with weakly projecting, poorly defined, narrow convexity laterally flanged by five short, spiniform setae.

Elytra. Length/width ratio 3.03–3.21; widest at anterior 1/6; anterior margins jointly 1.5–2× wider than posterior margin of pronotum; lateral margins nearly straight and sub-parallel after anterior 1/5, converging in posterior 1/3. Posterior declivity angled at 70–75° to main body axis. Elytra with 10 complete striae; striae broadly sculpted; punctures broad and faint beneath appressed scales, separated by 4–5× their diameter; intervals elevated.

Abdominal sterna. Ventrite III anteromesally incurved around a fovea located mesally on anterior margin, posterior margin elevated and set off from IV along lateral 3/8 s of its length. Sternum VII mesally 2/3× as long as wide; setae slightly lengthening, and becoming medially directed in posterior 1/3; anterior margin weakly curved; posterior margin distinctly incurved mesally, appearing broadly notched; surface of sternite concave, appearing broadly foveate, immediately anteriad of marginal incurvature.

Tergum. Tergum VII mesally incurved. Pygidium sub-cylindrical; medial 1/3 of anterior 2/3 of pygidium less sclerotized, with a patch of very short, fine setae.

Sternum VIII. Anterior laminar edges each incurved forming a 130° angle with lateral margin; slightly less sclerotized medially between arms; posterior margin medially incurved.

Ovipositor. Coxites as long as broad; styli with three setae near the base.

Spermatheca. ?-shaped; collum short, apically with a large, angulate, hood-shaped projection angled at 45° to corpus, sub-equal in length to ramus and continuously aligned with curvature of bulb of ramus; collum sub-contiguous with, and angled at ca. 60° to ramus; ramus basally elongate and constricted, forming a stalk, 1/3× length of collum, bulbous apically, 3× thicker than stalk; corpus not swollen, of equal thickness to collum and cornu; cornu elongate, apically, gradually narrowed, strongly recurved in basal 1/3, straight along mesal 1/3, and curved near apical 1/3 such that apex is parallel to collum and corpus.

Description of male

Not available or known.

Etymology. Named in reference to the short, apically explanate setae interspersed throughout the dorsum, which give this species a distinctly “knobbed” appearance; tylotos—knobby; Greek adjective (Brown, 1956).

Material examined

Holotype. ♀ “H. O. Canyon,; Davis Mts., Texas; Jeff Davis County; VII-20-1968, 6200; J. E. Hafernik” (TAMU).

Paratypes. “24 mi. wsw. Ft. Davis; Jeff Davis Co., Texas; August 17, 1969; Board & Hafernik” (TAMU: 1 ♀); “USA Texas Jeff Davis Co.; 4.1 mi. S. Fort Davis; sweeping grasses-weeds; 4750. 19.VII.82; R.S. Anderson” (CMNC: 1 ♀).

Distribution. This species has been found in three localities near the Davis Mountains in Jeff Davis County and in nearby Presidio County, Texas (USA). Habitat models (Fig. 5) predict that this represents the northeastern extent of its range, indicating a strong likelihood that it is present in other parts of the northern Chihuahuan desert, especially in the state of Chihuahua (México).

Natural history. No host plant associations have been documented. It is unknown whether this species is parthenogenetic.

Checklist of Species

RCC–5 articulations are provided in bold font. See Jansen & Franz (2015) for alignments of Minyomerus concepts published from 1831 to 2015.

Minyomerus Horn, 1876: 17 sec. Jansen & Franz (2018)

  == (INT) AND > (OST) Minyomerus Horn, 1876 sec. Jansen & Franz (2015)

  > AND = Elissa Casey, 1888: 271 sec. Casey (1888)

   (synonymized by Kissinger, 1964: 30)

  > AND = Pseudelissa Casey, 1888: 273 sec. Casey (1888)

   (synonymized by Pierce, 1909: 359)

  > AND = Piscatopus Sleeper, 1960: 84 sec. Sleeper (1960)

   (synonymized by Jansen & Franz, 2015: 12)

 microps (Say, 1831: 9) sec. Jansen & Franz (2015) [redescribed, p. 45]

  == (INT) AND > (OST) AND = Minyomerus innocuus Horn, 1876: 18 sec. Horn (1876)

   [former type of Minyomerus, designated by Pierce, 1913: 400]

   (synonymized by Jansen & Franz, 2015: 45)

  == (INT) AND > (OST) AND = Thylacites microps Say, 1831: 9 sec. Say (1831)

   (transferred to Minyomerus on the authority of Buchanan in litt.

   by Blackwelder & Blackwelder, 1948: 46)

  == (INT) AND > (OST) AND = Thylacites microsus Boheman, 1833: 523 sec. Boheman (1833)

   (synonymized by LeConte, 1859: 286)

 aeriballux Jansen & Franz, 2015: 52 sec. Jansen & Franz (2015)

 ampullaceus sp. nov. sec. Jansen & Franz (2018)

 bulbifrons Jansen & Franz, 2015: 81 sec. Jansen & Franz (2015)

 caseyi (Sharp, 1891: 151) sec. Jansen & Franz (2015) [redescribed, p. 66]

  == AND = Pseudelissa caseyi Sharp, 1891: 151 sec. Sharp (1891)

   (generic name synonymized by Pierce, 1909: 359)

 conicollis Green, 1920: 194 sec. Jansen & Franz (2015) [redescribed, p. 33]

 constrictus (Casey, 1888: 272) sec. Jansen & Franz (2015) [redescribed, p. 22]

  == AND = Elissa constricta Casey, 1888: 272 sec. Casey (1888)

   (generic name synonymized by Kissinger, 1964: 30)

 cracens Jansen & Franz, 2015: 61 sec. Jansen & Franz (2015)

 franko sp. nov. sec. Jansen & Franz (2018)

 gravivultus Jansen & Franz, 2015: 92 sec. Jansen & Franz (2015)

 griseus (Sleeper, 1960: 84) sec. Jansen & Franz (2015) [redescribed, p. 96]

  == AND = Piscatopus griseus Sleeper, 1960: 84 sec. Sleeper (1960)

   (generic name synonymized by Jansen & Franz, 2015: 96)

 imberbus Jansen & Franz, 2015: 18 sec. Jansen & Franz (2015)

 languidus Horn, 1876: 18 sec. Jansen & Franz (2015) [redescribed, p. 40]

  == (INT) AND > (OST) Minyomerus languidus Horn, 1876: 18 sec. Horn (1876)

  == AND = Pseudelissa cinerea Casey, 1888: 274 sec. Casey (1888)

   (synonymized by Pierce, 1909: 359)

 laticeps (Casey, 1888: 272) sec. Jansen & Franz (2015) [redescribed, p. 27]

  == AND = Elissa laticeps Casey, 1888: 272 sec. Casey (1888)

   (generic name synonymized by Kissinger, 1964: 30)

 politus Jansen & Franz, 2015: 86 sec. Jansen & Franz (2015)

 puticulatus Jansen & Franz, 2015: 75 sec. Jansen & Franz (2015)

 reburrus Jansen & Franz, 2015: 57 sec. Jansen & Franz (2015)

 rutellirostris Jansen & Franz, 2015: 103 sec. Jansen & Franz (2015)

 sculptilis sp. nov. sec. Jansen & Franz (2018)

 trisetosus Jansen & Franz, 2015: 71 sec. Jansen & Franz (2015)

 tylotos sp. nov. sec. Jansen & Franz (2018)

Species Identification Key

1 Procoxae apparently separate, with intercoxal processes touching or very nearly so2

–  Procoxae apparently contiguous, with intercoxal processes short and not touching3

2 (1) Rostrum approximately square and as wide as head in dorsal view; ramus of spermatheca basally narrow, forming a stalk that tapers into an apical bulbMinyomerus rutellirostris [JF2015]

– Rostrum approximately trapezoidal and narrower than the head in dorsal view; ramus of spermatheca cylindrical, somewhat bulbous, and basally constrictedMinyomerus griseus [JF2015]

3 (1) Anterior margin of pronotum bearing a full, well-developed tuft of 10 or more ocular vibrissae; anterolateral margins of prementum explanate, angular, and posteriorly declivous, with a distinctly hexagonal appearance4

– Ocular vibrissae reduced in number or length; anterior margins of prementum not explanate and declivous, typically with a pentagonal appearance5

4 (3) Head very wide and only somewhat swollen between eyes; rostrum ca. 4× wider than long in dorsal view; pronotum in dorsal view cylindrical; elytral setae short, brown, and sub-recumbent; ramus of spermatheca stalked and with apical bulb abruptly constricted, not tapering at point of connection to stalkMinyomerus laticeps [JF2015]

– Head and rostrum typical (rostrum 2–3× wider than long in dorsal view); pronotum in dorsal view somewhat globular, with a strong anterior constriction; elytral setae short and setiform, especially near disk; spermatheca without basal stalkMinyomerus constrictus [JF2015]

5 (3) Metatibial apex strongly convex, with setae similar in length to those of remainder of leg, somewhat lighter in color and translucent, and slightly lamelliform; head somewhat conical in form, rounded between the eyes; elytral setae copious, not in uniform rows on intervals, instead appearing in offset rows, especially near elytral suture and declivity6

– Metatibial apex oblique or weakly convex, with setae short and conical in appearance; head roughly quadrate; elytral setae in relatively uniform rows on elytra, not strongly offset7

6 (5) Elytral striae deeply and distinctly punctate, appearing pin-striped; elytra without obvious humeri, gradually widening posteriorly; ramus of spermatheca elongate, annulate, and sub-apically situated on corpusMinyomerus aeriballux [JF2015]

– Elytral striae punctate, with punctures somewhat obscured by appressed scales; elytra somewhat pyriform, with weak, but obviously present humeri; ramus of spermatheca elongate, somewhat swollen, and sub-apically situated on corpusMinyomerus reburrus [JF2015]

7 (5) Elytra very strongly convex in lateral view; anterior margin of pronotum wider than posterior margin; spermatheca comma-shaped, with ramus reduced, apically flattened and sub-contiguous with the collum; aedeagal pedon membranous ventrally, and not fully sclerotizedMinyomerus conicollis [JF2015]

– Elytra only somewhat convex to nearly flat in lateral view; anterior margin of pronotum similar in length to posterior margin; spermatheca variable; aedeagal pedon fully sclerotized8

8 (7) Body shape distinctly flask-like, with strongly constricted, sub-cylindrical pronotum and greatly protuberant elytra; in dorsal view, elytra nearly 2× width of pronotum at maximum width and nearly 3/4× as wide as long; in lateral view, anterior and posterior declivities of elytra abrupt and nearly vertical, with anterior elytral margin projecting strongly and characteristically dorsad of articulation with posterior pronotal margin; corpus of spermatheca with highly elongate projection aligned with midline of the ramus, which is basally tapered and angled at nearly 45° to corpusMinyomerus ampullaceus [JF2018], sp. n.

– Body shape usually narrow; elytra typically not more than 1.5× width of pronotum and typically not more than 2/3× as wide as long in dorsal view; elytral declivities in lateral view variable, but anterior margin never abruptly and strongly projected dorsad of posterior pronotal margin; spermatheca variable, but never with elongate projection aligned with midline of ramus9

9 (8) Setae of elytral disc a mix of shorter, brown setae and longer, more erect, white setae10

– Setae of elytral disc uniform12

10 (9) Anterior margin of pronotum bearing strikingly long setae, which project laterally up to 80° from longitudinal body axis and at least equal to diameter of eye; spermatheca with short, somewhat bulbous corpus, ramus sub-equal in size and perpendicular to corpus, and collum strongly recurved along basal 1/3 of its length; aedeagal pedon relatively short and wide, and abruptly constricted in apical 1/5, thereafter tapered to rounded pointMinyomerus franko [JF2018], sp. n.

– Anterior margin of pronotum bearing setae more strongly directed anteriorly and never as long as diameter of eye; spermatheca variable; aedeagal pedon, where known, narrow and expanded laterally in region of ostium11

11 (10) Setae apically explanate, appearing somewhat spatulate; corpus of spermatheca uniquely elongate, ramus short and cylindricalMinyomerus caseyi [JF2015]

– Setae linear; corpus of spermatheca typical, ramus bulbous and basally constrictedMinyomerus trisetosus [JF2015]

12 (9) Anterior margin of pronotum lined with linear setae that extend anteriorly beyond margin by half their length13

– Anterior margin of pronotum lacking setae, or with setae that do not extend far beyond margin14

13 (12) Lateral margins of gular cavity strongly rounded, never straight, and slightly longer than posterior margin; frons weakly projected between eyes; appressed scales on elytra without opalescent sheen; nasal plate with or without metallic reflections; lamina of spiculum ventrale sclerotized throughoutMinyomerus languidus [JF2015]

– Lateral margins of gular cavity nearly straight, and not longer than posterior margin; frons strongly projected between eyes; appressed scales with strong opalescent sheen; nasal plate with metallic reflections; lamina of spiculum ventrale with a membranous region present medially between laminar armMinyomerus gravivultus [JF2015]

14 (12) Elytra each 4–5× as long as broad in dorsal view, strongly punctate; elytra constricted anteriad of humeri, narrower than the pronotum, widening thereafter near the humeri; spermatheca with the corpus somewhat bulbous, and the ramus either flattened somewhat or slightly elongateMinyomerus cracens [JF2015]

– Elytra not so elongate, variably punctate; elytra lacking basal constriction; spermatheca variable15

15 (14) Elytral striae with large, obvious punctures16

– Elytral striae without evident punctures17

16 (15) Frons strongly protuberant; elytra in lateral view convex dorsally; spermatheca with corpus possessing an annulate, rectate projection nearly 1/2× length of ramus; aedeagal pedon evenly curving toward apex; aedeagal flagellum with spiriform apical sclerite that spirals counterclockwise and of equal length to aedeagal pedonMinyomerus bulbifrons [JF2015]

– Frons not so protuberant; elytra in lateral view nearly flat dorsally; spermatheca with corpus possessing an annulate, rectate projection nearly 2/3× length of the ramus; aedeagal pedon narrow and elongate; aedeagal flagellum with very minute apical scleriteMinyomerus puticulatus [JF2015]

17 (15) Frons strongly protruding in lateral view by ca. 2× diameter of eye18

– Frons not or weakly protruding in lateral view by 1.5× diameter of eye or less19

18 (17) Nasal plate defined by inversely V-shaped, impressed lines; spermatheca with the ramus elongate and apically swollen, corpus possessing an annulate, rectate projection nearly 1/2× length of the ramus, and cornu evenly recurved throughout its length; aedeagal flagellum with a spiriform apical sclerite that spirals clockwise and of equal length to pedonMinyomerus politus [JF2015]

– Nasal plate lacking distinct impressions; spermatheca with ramus basally tapered with a short stalk, corpus narrow and lacking an annulate rectate bulb, and cornu with an abrupt apical curve; males not knownMinyomerus tylotos [JF2018], sp. n.

19 (17) Ventrolateral sulci weakly defined as a notch ventrad of antennal insertion, or absent entirely; intervals broadly sculpted and raised, and striae not punctate; body generally robust in overall quality; appressed scales uniformly beige and gray, with a distinctly “crusty” appearance; spermatheca with ramus and collum appearing as two subcontiguous, apically invaginated bulbsMinyomerus microps [JF2015]

– Ventrolateral sulci deeply and distinctly defined along their entire length; intervals, if raised, only sculpted along lateral faces of elytra, not on disk; body usually not markedly robust; appressed scales either translucent or otherwise typical of genus, not beige and crusted; spermatheca distinctly sinuate, with well defined, protruding ramus20

20 (19) Elytra with very minute setae, only perceptible at high magnification; lateral faces of elytra with intervals not noticeably raised; ramus of spermatheca elongate, cylindrical, and slightly thinner than corpus, cornu strongly recurved in basal half with uniquely sinuate appearance, both corpus and cornu with hood-like projections shorter than ramus; males not knownMinyomerus imberbus [JF2015]

– Elytra with easily visible, linear setae; lateral faces of elytra with intervals raised; ramus of spermatheca bulbous, basally tapered, and similar in width to corpus, cornu strongly recurved, but short in basal half with uniquely sinuate appearance, both corpus and cornu with hood-like projections longer than ramus; aedeagal pedon narrow and cylindrical, apically taperedMinyomerus sculptilis [JF2018], sp. n.

Phylogenetic Results

A matrix of 52 characters was assembled for the 26 terminal taxa (Table 1). These characters are comprised of all 46 characters included in the revision of Minyomerus [JF2015], plus an additional six characters intended to identify putative sister taxa to the newly described species. Parsimony analysis returned a single, most-parsimonious cladogram (henceforth MPT) with a length (L) of 99 steps, a consistency index (CI) of 60 and a retention index (RI) of 80 (Farris, 1989); see Figs. 37–38. Tree analysis using New Technology was used to confirm that the shortest tree had been found (Goloboff, Farris & Nixon, 2008). The most-parsimonious cladogram is shown in Fig. 37, with relative and absolute bsv (see also Materials and Methods: Phylogenetic analysis) mapped along the left side of each branch; nodes with bootstrap support above 0.95 are marked with a “*” symbol to the right of each node. In a complementary graph, we show the herein used clade concept labels (Fig. 38).

Figure 37 Preferred phylogeny—character transitions and support.

Single most-parsimonious cladogram representing the preferred phylogeny of species of Minyomerus [JF2018], and select outgroup taxa (L = 99, CI = 60, RI = 80). Characters 9, 27, 39, 45–47, 49, and 51 are mapped under ACCTRAN optimization; all others are unambiguously optimized. Black squares indicate non-homoplasious character state changes, whereas white squares indicate homoplasious character state changes. The numbers above and below the squares represent character numbers and states, respectively. Bremer support (upper value) and rfd (lower value) values can be found at the left ends of the branches. A “*” symbol at the right end of a branch indicates Bootstrap support greater than 0.95.

Figure 38 Preferred phylogeny—clade concept labels.

Topology and species-level taxonomic concept labels as in Fig. 37. Clade concept labels, numbered 1–20, are consistently generated by using the alphabetically first epithet in each of the bifurcating sister clades. This method safeguards the clade concept labels against changes due simply to reorientation of leaves. Bold-font square brackets indicate new [JF2018] labels. See also RCC–5 Alignments.

Table 1 Taxon/character matrix used for cladistic analysis.

Taxon\character	0	1	1	2	2	3	3	4	4	5	
5	0	5	0	5	0	5	0	5	0	
Sitona californicus [non-focal]	00-00	?????	00000	00000	00000	00000	00—	–0—	–???	????? ??	
Pandeleteius cinereus [non-focal]	11000	?????	01000	00001	01000	00100	00000	000–0	00???	????? ??	
Pandeleteinus subcancer [non-focal]	11000	?????	01000	00001	01010	00100	00000	000–0	00???	????? ??	
Isodrusus debilis [non-focal]	11000	?????	01000	00001	01011	00100	00000	000–0	00???	????? ??	
Isodacrys buchanani [non-focal]	11000	?????	01000	00001	01011	00101	00000	000–0	00???	????? ??	
Minyomerus constrictus [JF2015]	21100	00010	02110	01002	00011	11211	00000	00000	00000	01010 00	
Minyomerus laticeps [JF2015]	21100	00010	02110	01002	00011	11211	00000	00000	00000	01010 00	
Minyomerus imberbus [JF2015]	21100	?????	02010	11002	10011	11211	01001	00000	00???	????? ??	
Minyomerus sculptilis [JF2018]	21100	?????	02010	11002	10011	11211	01001	00000	00001	00000 10	
Minyomerus conicollis [JF2015]	21100	00000	02010	11002	10011	10211	00001	10000	01000	00000 00	
Minyomerus languidus [JF2015]	21000	11100	02010	11002	10011	10211	00001	10000	?????	????? ??	
Minyomerus microps [JF2015]	21001	11101	02110	11002	10011	10211	00001	10000	10???	????? ??	
Minyomerus tylotos [JF2018]	21001	11101	02110	11002	10011	10211	00001	10000	00???	????? ??	
Minyomerus cracens [JF2015]	21000	11101	02020	11112	10011	11211	00001	10001	00000	10010 10	
Minyomerus ampullaceus [JF2018]	21000	?????	02020	11??2	10011	11211	00001	11001	00???	????? ??	
Minyomerus aeriballux [JF2015]	22000	11101	12020	11012	10011	20211	00001	11001	10000	00000 01	
Minyomerus reburrus [JF2015]	22000	11101	12020	11112	10011	20211	00001	11021	00???	????? ??	
Minyomerus franko [JF2018]	21110	10100	02020	11002	10011	11211	00001	10011	10010	00000 01	
Minyomerus caseyi [JF2015]	21110	00101	02020	11112	10011	11211	00001	10101	10010	10010 01	
Minyomerus trisetosus [JF2015]	21110	00101	02020	11012	10011	10211	00001	10101	10???	????? ??	
Minyomerus gravivultus [JF2015]	21100	11101	02120	11002	10011	10211	00111	10000	??010	00000 11	
Minyomerus griseus [JF2015]	21100	10101	02120	01002	10111	11211	00111	10000	00010	01100 10	
Minyomerus rutellirostris [JF2015]	21100	10101	02120	11002	10111	11211	00111	10000	00010	01100 10	
Minyomerus puticulatus [JF2015]	21000	11101	02020	11012	10011	11211	10101	10010	01011	01000 11	
Minyomerus bulbifrons [JF2015]	21000	11101	02021	11112	10011	10211	10101	10000	01110	01001 01	
Minyomerus politus [JF2015]	21000	?????	02021	11112	10011	10211	10101	10010	01111	01001 11	
Notes:

Includes all species of Minyomerus [JF2015], newly designated species, and select outgroup taxa. All multi-state characters coded as additive, except for character 33. The symbol “–” denotes inapplicable character states, whereas “?” denotes missing information (see also text).

The characters, states, and preferred optimizations are described in this section. Characters relating to placement of the herein described taxa are discussed in detail in the Discussion section, along with changes in species-group composition and tree topology from Jansen & Franz (2015). For all characters not resolved as unreversed synapomorphies, both the individual consistency (ci) and retention (ri) indices are provided.

Habitus, form of appressed scales: (0) elongate pyriform, not overlapping; (1) sub-circular to polygonal, variously overlapping non-linearly; (2) sub-circular and only overlapping posteriorly. Coded as additive due to alignment of character states with the preferred phylogeny. Coding as non-additive in isolation or in unison with other additive multi-state characters does not affect polarization of the character/states or alter the phylogeny. State 1 is a synapomorphy for the tanymecine clade [non-focal], whereas state 2 is a synapomorphy for Minyomerus [JF2018].

Habitus, arrangement of elytral setae: (0) variously interspersed; (1) arranged in single-file rows on elytral intervals; (2) arranged non-uniformly on elytral intervals. Coded as additive due to alignment of character states with the preferred phylogeny. Coding as non-additive in isolation or in unison with other additive multi-state characters does not affect polarization of the character/states or alter the phylogeny. State 1 is a synapomorphy for the tanymecine clade [non-focal], whereas state 2 is a synapomorphy the M. aeriballux–M. reburrus clade [JF2015].

Habitus, lateral elytral setae and ventral setae differentiated from setae of elytral disc: (0) absent; (1) present. Homoplasy for Minyomerus [JF2018], with a reversal (state 0) in the M. aeriballux–M. languidus clade [JF2015], subsequent convergent gain (state 1) in the M. bulbifrons–M. caseyi clade [JF2018], and convergent reversal (state 0) in the M. bulbifrons–M. puticalutus clade [JF2015] (ci = 25; ri = 70).

Habitus, rows of elytral setae with larger white setae randomly interspersed among smaller brown setae: (0) absent; (1) present. Synapomorphy for the M. caseyi–M. franko clade [JF2018]. Changed from Jansen & Franz (2015), where M. rutellirostris [JF2015] was previously coded as having this character; however, the white elytral setae of this species are not randomly interspersed, but follow a distinct, and uniquely derived, pattern where every other interval contains a row of such setae.

Habitus, elytra and pronotum generally large, protuberant, and sculpted in appearance along dorsal and lateral faces: (0) absent; (1) present. Synapomorphy for the M. microps–M. tylotos clade [JF2018].

Prementum, anterior margin medially with a distinct facet, rather than a single edge, that continues to lateral margins: (0) absent; (1) present. Synapomorphy for the M. aeriballux–M. languidus clade [JF2015], with a single reversal in the M. caseyi–M. trisetosus clade [JF2015] (ci = 50; ri = 75).

Prementum, strongly ligulate and with margins nearly straight, appearing pentagonal: (0) absent; (1) present. Synapomorphy for the M. aeriballux–M. languidus clade [JF2015], with independent reversals in the M. caseyi–M. franko clade [JF2018] and M. griseus–M. rutellirostris clade [JF2015], respectively (ci = 33; ri = 71).

Prementum, anterolateral margins simple, unexpanded: (0) absent; (1) present. Synapomorphy for the M. aeriballux–M. languidus clade [JF2015].

Prementum, anterolateral margins explanate, angular, and posteriorly declivous, with a distinctly hexagonal appearance: (0) absent; (1) present. ACCTRAN optimization preferred (see Agnarsson & Miller, 2008), therefore inferred as a synapomorphy for the M. constrictus–M. laticeps clade [JF2015].

Prementum, exposure of palpomere I: (0) exposed, visible beyond ligula and anterior margin of prementum in ventral view; (1) hidden, fully covered or only minutely exposed beyond ligula and anterior margin of prementum in ventral view. Synapomorphy for the M. aeriballux–M. microps clade [JF2015], with a single reversal in M. franko [JF2018] (ci = 50; ri = 75).

Rostrum, form in dorsal view: (0) approximately quadrate; (1) somewhat conical, medially convex. Synapomorphy for the M. aeriballux–M. reburrus clade [JF2015].

Rostrum, form of nasal plate and demarcation of epistoma: (0) with three parallel, longitudinal carinae, and surface planar between these; (1) with a sharp, narrow, chevron-shaped carina demarcating epistoma; (2) with a broad, scale-covered, chevron-shaped carina demarcating epistoma. Coded as additive due to alignment of character states with preferred phylogeny. Coding as non-additive in isolation or in unison with other additive multi-state characters does not affect polarization of the character/states or alter the phylogeny. State 1 is a synapomorphy for the tanymecine clade [non-focal], whereas state 2 is a synapomorphy for Minyomerus [JF2018].

Rostrum, sulcus posteriad of nasal plate weakly impressed: (0) absent; (1) present. Convergently present in the M. constrictus–M. laticeps clade [JF2015], the M. microps–M. tylotos clade [JF2018], and the M. gravivultus–M. griseus clade [JF2015] (ci = 33; ri = 60).

Rostrum, form of sulcus posteriad of nasal plate: (0) absent; (1) sulcus present, broad, and weakly punctate; (2) sulcus present, more strongly punctate. Coded as additive due to alignment of character states with preferred phylogeny. Coding as non-additive in isolation or in unison with other additive multi-state characters does not affect polarization of the character/states or alter the phylogeny. Synapomorphy for Minyomerus [JF2018] (state 1) and the M. aeriballux–M. cracens clade [JF2015] (state 2), respectively.

Head, frons very strongly projected beyond anterior margin of eye, by 2× anterior–posterior length of eye: (0) absent; (1) present. Synapomorphy for the M. bulbifrons–M. politus clade [JF2015].

Head, frons with posterior transverse constriction: (0) absent; (1) present. Synapomorphy for the M. aeriballux–M. languidus clade [JF2015], with a single reversal in M. griseus [JF2015] (ci = 50, ri = 85).

Antenna, length of scrobe relative to funicle and club: (0) scrobe shorter than funicle and club combined; (1) scrobe subequal in length to funicle and club combined. Synapomorphy for Minyomerus [JF2018].

Antenna, terminal funicular segment entirely without thin, nearly setiform scales: (0) absent; (1) present. Convergently present in M. cracens [JF2015], M. reburrus [JF2015], M. caseyi [JF2015], and the M. bulbifrons–M. politus clade [JF2015] (ci = 25; ri = 25).

Antenna, terminal funicular segment at least partially clothed with broad scales: (0) absent; (1) present. Synapomorphy for the M. aeriballux–M. cracens clade [JF2018] with independent reversals in M. franko [JF2018] and the M. gravivultus–M. griseus clade [JF2015] (ci = 33; ri = 71).

Head, angle of base in relation to prothorax: (0) directed anteriorly, in line with main body axis; (1) directed strongly ventrally; (2) directed slightly ventrally. Coded as additive due to alignment of character states with preferred phylogeny. Coding as non-additive in isolation or in unison with other additive multi-state characters does not affect polarization of the character/states or alter the phylogeny. State 1 is a synapomorphy for the tanymecine clade [non-focal], whereas state 2 is a synapomorphy for Minyomerus [JF2018].

Pronotum, condition of post-ocular vibrissae: (0) present in a well-developed tuft of 10 or more setae; (1) present in a reduced tuft of 3–7 setae. Synapomorphy for the M. aeriballux–M. imberbus clade [JF2018].

Prosternum, intercoxal process complete, undivided: (0) absent; (1) present. Synapomorphy for the tanymecine clade [non-focal], with a single reversal for Minyomerus [JF2018] (ci = 50; ri = 66).

Prosternum, intercoxal process divided at midpoint between coxae, but both anterior and posterior processes extending completely between procoxae and contiguous with each other: (0) absent; (1) present. Synapomorphy for the M. griseus–M. rutellirostris clade [JF2015].

Legs, fore femora not swollen in comparison to other legs: (0) absent; (1) present. Synapomorphy for the M. aeriballux–P. subcancer clade [non-focal].

Legs, sculpture of ventral surface of protibiae: (0) evenly convex throughout; (1) with a longitudinal groove or concavity. Synapomorphy for the M. aeriballux–I. debilis clade [non-focal].

Legs, setation of metatibial apex: (0) bristles at least as long as surrounding setae and setiform; (1) bristles shorter than surrounding setae and conical; (2) bristles sub-equal in length to surrounding setae and somewhat lamelliform. Coded as additive due to alignment of character states with preferred phylogeny, and the appearance of being a transformation series. Coding as non-additive in isolation or in unison with other additive multi-state characters does not affect polarization of the character/states or alter the phylogeny. Synapomorphy for Minyomerus [JF2018] (state 1) and the M. aeriballux–M. reburrus clade [JF2015] (state 2), respectively.

Legs, curvature of metatibial apex: (0) convex; (1) oblique. ACCTRAN optimization preferred (see Agnarsson & Miller, 2008), therefore inferred as a synapomorphy for Minyomerus [JF2018] with a reversal (state 0) in the M. aeriballux–M. conicollis clade [JF2015], then a convergent gain (state 1) in the M. aeriballux–M. bulbifrons clade [JF2018], with independent reversals (state 0) in the M. aeriballux–M. reburrus clade [JF2015], M. gravivultus [JF2015], M. trisetosus [JF2015], and the M. bulbifrons–M. politus clade [JF2015] (ci = 14; ri = 40).

Legs, relative length of mesotarsi to mesotibiae: (0) tarsi less than 3/4× length of tibiae; (1) tarsi at least equal in length to tibiae; (2) tarsi shorter than tibiae, but longer than 3/4× length of tibiae. Coded as additive due to alignment of character states with preferred phylogeny. Coding as non-additive in isolation or in unison with other additive multi-state characters does not affect polarization of the character/states or alter the phylogeny. State 1 is a synapomorphy for the tanymecine clade [non-focal], whereas state 2 is a synapomorphy for Minyomerus [JF2018].

Legs, tarsi ventrally spinose: (0) absent; (1) present. Synapomorphy for Minyomerus [JF2018].

Elytra, humeral angle rounded, not projected: (0) absent; (1) present. Synapomorphy for the M. aeriballux–I. buchanani clade [non-focal].

Female terminalia, spermatheca with apical cylindrical bulb on corpus: (0) absent; (1) present. Synapomorphy for the M. bulbifrons–M. puticulatus clade [JF2015].

Female terminalia, corpus of spermatheca sinuate: (0) absent; (1) present. Synapomorphy for the M. imberbus–M. sculptilis clade [JF2018].

Female terminalia, lamina of spiculum ventrale less sclerotized between laminar arms: (0) absent; (1) present. Coded as inapplicable for S. californicus [non-focal], as laminar arms are not apparent. Synapomorphy for the M. gravivultus–M. griseus clade [JF2015].

Female terminalia, lamina of spiculum ventrale with laminar arms bifurcating around a membranous region: (0) absent; (1) present. Coded as inapplicable for S. californicus [non-focal], as laminar arms are not apparent. Synapomorphy for the M. gravivultus–M. griseus clade [JF2015].

Female terminalia, lamina of spiculum ventrale with style basally divided or obscured, not mesally intact: (0) absent; (1) present. Coded as inapplicable for S. californicus [non-focal], as laminar arms are not apparent. Synapomorphy for the M. aeriballux–M. imberbus clade [JF2015].

Female terminalia, lamina of spiculum ventrale with laminar arms clearly bifurcating. (0) absent; (1) present. Coded as inapplicable for S. californicus [non-focal], as laminar arms are not apparent. Synapomorphy for the M. aeriballux–M. conicollis clade [JF2015].

Female terminalia, laminar arms narrowly bifurcating basally, thereafter sub-parallel mesally: (0) absent; (1) present. Synapomorphy for the M. aeriballux–M. ampullaceus clade [JF2018].

Female terminalia, coxites of ovipositor with a lateral, anteriorly-directed, recurved, alate process: (0) absent; (1) present. Coded as inapplicable for S. californicus [non-focal], as coxites of ovipositor are not apparent. Synapomorphy for the M. caseyi–M. trisetosus clade [JF2015].

Female terminalia, relative length of styli to coxites of ovipositor: (0) Similar in size; (1) distinctly shortened; (2) highly reduced, appearing minute. Coded as non-additive, due to strong differences in structure of coxites and styli in state 2; inapplicable for outgroup taxa, as styli of ovipositor are not apparent. ACCTRAN optimization preferred (see Agnarsson & Miller, 2008), therefore inferred as convergent gains in M. franko [JF2018] and the M. bulbifrons–M. puticulatus clade [JF2015] (state 1), with a single reversal in M. bulbifrons [JF2015] (state 0). Autapomorphy for M. reburrus [JF2015] (state 2) (ci = 50, ri = 0).

Female terminalia, condition of medial, anteriorly-directed, sclerotized process of coxites of ovipositor: (0) fully developed; (1) reduced and inapparent. Coded as inapplicable for S. californicus [non-focal], as coxites of ovipositor are not apparent. Synapomorphy for the M. aeriballux–M. cracens clade [JF2015], with a single reversal in the M. gravivultus–M. griseus clade [JF2015] (ci = 50, ri = 83).

Female terminalia, anterior margin of tergum VII entirely free of sclerotized band: (0) absent; (1) present. Coded as inapplicable for S. californicus [non-focal], as the tergum VII is evenly sclerotized throughout. Convergently present in M. aeriballux [JF2015], M. microps [JF2015], and the M. caseyi–M. trisetosus clade [JF2018] (ci = 33; ri = 50).

Female terminalia, anterior margin of tergum VII sclerotized fully, appearing as an obviously complete band: (0) absent; (1) present. Coded as inapplicable for S. californicus [non-focal], as the tergum VII is evenly sclerotized throughout. Convergently present in M. conicollis [JF2015] and the M. bulbifrons–M. puticulatus clade [JF2015] (ci = 50; ri = 66).

Male terminalia, apical sclerite of aedeagal flagellum elongate-spiriform: (0) absent; (1) present. Synapomorphy for the M. bulbifrons–M. politus clade [JF2015].

Male terminalia, style of spiculum gastrale with an anterior ventral flange: (0) absent; (1) present. Synapomorphy for the M. bulbifrons–M. caseyi clade [JF2018].

Male terminalia, lamina of spiculum gastrale longer than broad and anteriorly extended along style: (0) absent; (1) present. ACCTRAN optimization preferred (see Agnarsson & Miller, 2008), therefore inferred as convergent gains in the M. imberbus–M. sculptilis clade [JF2018] and the M. bulbifrons–M. puticulatus clade [JF2015], with a reversal in M. bulbifrons [JF2015] (ci = 33; ri = 0).

Male terminalia, sub-triangular sclerites of sternum VIII with a medial process: (0) absent; (1) present. ACCTRAN optimization preferred (see Agnarsson & Miller, 2008), therefore inferred as convergent gains in M. cracens [JF2015] and the M. caseyi–textit M. trisetosus clade [JF2015] (ci = 50, ri = 0).

Male terminalia, curvature of posterior margin of tergum VII: (0) evenly arcuate; (1) medially incurved. ACCTRAN optimization preferred (see Agnarsson & Miller, 2008), therefore convergently present in the M. constrictus–M. laticeps clade [JF2015] and the M. bulbifrons–M. gravivultus clade [JF2015] with a reversal in M. gravivultus [JF2015] (ci = 33; ri = 66).

Male terminalia, tergum VII approximately 4× as long as broad: (0) absent; (1) present. Synapomorphy for the M. griseus–M. rutellirostris clade [JF2015].

Male terminalia, aedeagal pedon expanded laterally around ostium: (0) absent; (1) present. ACCTRAN optimization preferred (see Agnarsson & Miller, 2008), therefore convergently present in the M. constrictus–M. laticeps clade [JF2015], M. cracens [JF2015], and the M. caseyi–M. trisetosus clade [JF2015] (ci = 33; ri = 33).

Male terminalia, aedeagal pedon broad basally, evenly tapering toward apex: (0) absent; (1) present. Synapomorphy for the M. bulbifrons–M. politus clade [JF2015].

Male terminalia, aedeagal pedon medially sclerotized along dorsum: (0) absent; (1) present. ACCTRAN optimization preferred (see Agnarsson & Miller, 2008), therefore convergently present in the M. imberbus–M. sculptilis clade [JF2015], M. cracens [JF2015], and the M. bulbifrons–M. gravivultus clade [JF2015], with a reversal in M. bulbifrons [JF2015] (ci = 25; ri = 50).

Male terminalia, width of connection between apodemes of aedeagal tegmen: (0) wider than base of apodeme; (1) narrower than base of apodeme. Synapomorphy for the M. aeriballux–M. bulbifrons clade [JF2018], with a single reversal in the M. griseus–M. rutellirostris clade [JF201] (ci = 50; ri = 83).

RCC–5 Alignments

Details of our RCC–5 alignment approached are given in free text form in the Supplemental Information SI1, which also describes the content of the data input and output files. The latter, in turn, are appended in .txt, .csv, and .pdf format in the Supplemental Information SI2–SI4. All shown alignments are intensional in the sense of Franz & Peet (2009), and thus maximize high-level concept congruence where indicated, and in spite of non-congruent lower-level concept sampling.

The first, classification-based alignment (Fig. 1) is simple and straightforward to interpret (see also Supplemental Information SI2). We obtain high-level congruence among the concepts Minyomerus [JF2018] and Minyomerus [JF2015], where 17 species-level concepts are retained from Jansen & Franz (2015) and four species-level concepts are added in the current review. The coverage constraint is relaxed for Minyomerus [JF2015], thus allowing the four new species-level concepts to be subsumed under this parent. This is based on our assertion that they fall under the generic character circumscription of Jansen & Franz (2015).

The following two Figs. 2–3 show fully bifurcated, multi-phylogeny alignments of the same reasoner toolkit input, but resolved as whole concepts vs. split concepts, respectively. In Fig. 2 (Supplemental Information SI3), we observe that the phylogenetic placements of two of the four new species-level concepts cause significant non-congruence in the alignment, resulting in seven overlapping RCC–5 articulations. Minyomerus franko [JF2018] is subsumed under the M. caseyi–M. franko clade [JF2018], which is intensionally congruent with the M. caseyi–M. trisetosus clade [JF2015]. In other words, this placement is not the source of non-congruence in the alignment. Similarly, the placement of M. tylotos [JF2018] into the new M. microps–M. tylotos clade [JF2018] is not conflicting in an intensional sense. At the next, more inclusive level, this addition “resolves” into the congruent M. aeriballux–M. microps clade [JF2018]/[JF2015].

In contrast, the placement of M. ampullaceus [JF2018] “inside” of M. cracens [JF2015] in the current phylogeny, generates five overlapping articulations among as many (five) non-congruent concept regions positioned 1–2 levels above these species-level concepts. The conflict is resolved in the next, more inclusive and congruent region of the M. aeriballux–M. cracens clade [JF2018] == M. aeriballux–M. bulbifrons clade [JF2015].

The placements of the previously circumscribed M. imberbus [JF2015] and the new species-level concept M. sculptilis [JF2018]—in relation to the congruent clade M. constrictus–M. laticeps [JF2018]/[JF2015]—cause two additional instances of overlap (Fig. 2). In the current phylogeny, M. imberbus [JF2015] is sister to M. sculptilis [JF2018], and placed “inside” of the M. constrictus–M. laticeps clade [JF2018]/[JF2015]. However, in the preceding phylogeny sec. Jansen & Franz (2015), M. imberbus [JF2015] is non-congruently included in the M. constrictus–M. imberbus clade [JF2015]. This conflict is only resolved at the level of Minyomerus [JF2018]/[JF2015].

Figure 3 (Supplemental Information SI4) shows that the inclusion of the four new species-level concepts in the Minyomerus [JF2018] phylogeny generates five split-concept regions for which there are no adequate labels in either input phylogeny. These labels correspond to the overlapping articulations mentioned above; in particular the non-congruent assignments of M. ampullaceus [JF2018], M. cracens [JF2018], and M. sculptilis [JF2018]. The phylogenetic character evidence for these placements and relationships are discussed in the following sections.

Discussion

Relationships to the previous revision

The differences of the current phylogeny (Figs. 37–38) in relation to that of Jansen & Franz (2015) are in large part due to the unique character combinations present in the newly added species (Rieppel, 2007; Franz, 2014). Nonetheless, three main clades are resolved with strong support, and further corroborate the topology of Jansen & Franz (2015), as follows: Minyomerus [JF2018] is strongly supported by the same eight synapomorphies identified in Jansen & Franz (2015). These are reiterated in the Introduction (bsv [henceforth: bsv] = 10, rfd [henceforth: rfd] = 95; Bootstrap [henceforth: boot] = 100).

Minyomerus griseus [JF2015] forms a well-supported clade with M. rutellirostris [JF2015] (bsv = 4, rfd = 77, boot = 96). These taxa jointly share the same two synapomorphies (chars. 23:1 and 48:1) provided in Jansen & Franz (2015): (1) the intercoxal process is divided at the midpoint between the coxae, but has both the anterior and posterior processes extending completely between the procoxae and contiguous with each other; and (2) the male tergum VII is nearly 4× as long as broad, respectively. In addition, the M. gravivultus–M. griseus [JF2015] clade (bsv = 3, rfd = 60), as resolved in the current cladogram, is congruent with that of Jansen & Franz (2015).

Minyomerus [JF2018] is nested within a well-supported clade of Tanymecini [non-focal] (boot = 100). However, further work is needed to assess the phylogenetic relationships between all genera presently assigned to the Tanymecini [non-focal] (Alonso-Zarazaga & Lyal, 1999).

Intrageneric relationships

Within Minyomerus [JF2018], beginning at the earliest-bifurcating node and proceeding toward the leaves, the first major incongruence with Minyomerus [JF2015] is the placement of M. imberbus [JF2015]. This species was sister to the M. constrictus–M. laticeps [JF2015] clade, which in turn was sister to the M. aeruballux–M. conicollis clade [JF2015]. The present analysis places M. imberbus [JF2015] in a clade with M. sculptilis [JF2018] (see Placement of newly described species). The M. aeriballux–M. imberbus clade [JF2018] (bsf = 2, rfd = 50) is supported by three synapomorphies: (1) presence of a transverse constriction across the posterior of the frons (char. 16: 1); (2) presence of a reduced tuft of post-ocular vibrissae (char. 21: 1); and (3) a mesally obscure lamina of the spiculum ventrale in the female (char. 35: 1).

We resolve M. cracens [JF2015] as sister to the M. aeriballux–M. bulbifrons [JF2018] clade, inclusively supported by three synapomorphies: (1) presence of a strongly punctate sulcus posteriad of the nasal plate (char. 14: 2); (2) presence of broad scales on the terminal funicular segment of the antennae (char. 19: 1); and (3) absence of a medial, anteriorly-directed, sclerotized process on the coxites of the ovipositor (char. 40: 1).

The M. aeriballux–M. bulbifrons [JF2018] clade is weakly supported by a single synapomorphy: the width of the connection between the apodemes of the aedeagal tegmen is narrower than the base of the apodeme (char. 52: 1). Within this clade, the position of the M. bulbifrons–M. caseyi clade [JF2018] clade as separate from, and sister to, the M. aeriballux–M. ampullaceus clade [JF2018], is supported by one synapomorphy and one homoplasious character, namely: (1) presence of an anterior ventral flange on the style of the spiculum gastrale (char. 44: 1—synapomorphic), and (2) differentiation of the setae on the lateral portion of the elytra and on the venter from the setae on the elytral disc (char. 3: 1—homoplasious).

Placement of newly described species

Clades within Minyomerus [JF2018] not addressed in the preceding section are identical in topology and composition to those of Minyomerus [JF2015], except for the addition of newly described species. Here, we assess the phylogenetic placements of these species. We also discuss similarities in the biogeographic range of each species, in relation to the putative sister taxa, based on the results of species distribution modeling (see Figs. 4–7).

Minyomerus sculptilis

Myniomerus sculptilis [JF2018] is inferred as sister to M. imberbus [JF2015]. The M. imberbus–M. sculptilis clade [JF2018] (bsv = 3, rfd = 72) is supported by a single synapomorphy and two homoplasious characters: (1) corpus of spermatheca sinuate (char. 32: 1—synapomorphic); (2) lamina of spiculum gastrale in male longer than broad and anteriorly extended along style (char. 45: 1—homoplasious); and (3) aedeagal pedon medially sclerotized along dorsum (char. 51: 1—homoplasious). In addition to these characters, M. imberbus [JF2015] and M. sculptilis [JF2018] share a general external gestalt, which makes separating these two species difficult, especially in damaged or worn specimens.

Whereas M. sculptilis [JF2018] is associated with big sagebrush (Artemisia tridentata [non-focal], tumbleweed (Salsola tragus [non-focal], and tall tumblemustard (Sisymbrium altissimum [non-focal]; its sister taxon M. imberbus [JF2015] is associated with budsage (Artemisia spinescens [non-focal]. The divergence of these two species may have been driven in part by differences in host plant use. However, this is less likely considering the generalist feeding habits of Minyomerus [JF2018] congenerics. Conversely, their divergence may have resulted from a vicariance event, based on their present-day biogeographic distributions, which are separated by the eastern extension of the Columbia Plateau. Minyomerus sculptilis [JF2018] appears to be endemic to the Snake River Plain to the north, whereas M. imberbus [JF2015] has been found in the Great Basin Desert to the south.

Minyomerus tylotos

Minyomerus tylotos [JF2018] is sister to M. microps [JF2015]. The M. microps–M. tylotos clade [JF2018] (bsv = 3, rfd = 73) is supported by a single synapomorphy and a single homoplasious character: (1) elytra and pronotum generally large, protuberant, and sculpted in appearance along dorsal and lateral faces (char. 5: 1—synapomorphic); and (2) sulcus posteriad of nasal plate broad and weakly punctate (char. 13: 1—homoplasious). In addition to these characters, the two species share a similar gestalt and uniform setation.

Minyomerus tylotos [JF2018] appears to be endemic to northern Chihuahuan Desert, whereas M. microps [JF2015] is widely distributed to the north throughout the Great Plains and along the Missouri River. We consider it likely that M. microps [JF2015] represents a northern radiation of the common ancestor of this clade. Conversely, M. tylotos [JF2018] may represent the ancestral distribution to the south, based on the hypothesized origin of Minyomerus [JF2018] in the Chihuahuan Desert; see Jansen & Franz (2015) and Wilson & Pitts (2010).

Minyomerus ampullaceus

Minyomerus ampullaceus [JF2018] is sister to the M. aeriballux–M. reburrus clade [JF2015]. The M. aeriballux–M. ampullaceus clade [JF2018] (bsv = 1, rfd = 50) is supported by a single synapomorphy: lamina of spiculum ventrale with laminar arms basally bifurcating and sub-parallel mesally thereafter (char. 37: 1). The placement of this species is tentative and based on the characteristics of a single, worn specimen.

Nonetheless, the biogeographic distributions of the species in the M. aeriballux–M. ampullaceus clade [JF2018] exhibit overlap. Minyomerus ampullaceus [JF2018] is documented from Carlsbad, New Mexico, in the western parts of the distributions of M. aeriballux [JF2015] and M. reburrus [JF2015]. The divergence of the latter two species is thought to be a result of their habitat and host plant preference, given their overlapping ranges. Minyomerus aeriballux [JF2015] is found in very sandy soils and on dune systems, whereas M. reburrus [JF2015] prefers arid grasslands. Without additional distributional or host plant data for M. ampullaceus [JF2018], we cannot assess whether the single documented locality for this species represents the center or edge of its range. However, this locality does overlap with the known range of its sister clade, suggesting that the divergence of M. ampullaceus [JF2018] from the M. aeriballux–M. ampullaceus clade [JF2018] was not a vicariance event.

Minyomerus franko

Minyomerus franko [JF2018] is sister to the M. caseyi–M. trisetosus clade [JF2015]. The M. caseyi–M. franko clade [JF2018] (bsv = 4, rfd = 63) is supported by a single synapomorphy and two homoplasious characters: (1) rows of setae on elytral intervals comprised of larger white setae randomly interspersed among smaller brown setae(char. 4: 1—synapomorphic); (2) prementum lacking strong ligula and straight margins, not appearing pentagonal (char. 7: 0—homoplasious); and (3) anterior margin of female tergum VII entirely free of sclerotized band (char. 41: 1—homoplasious). In addition to these characters, members of this clade share a generally similar gestalt, especially regarding the head and rostrum, and the articulation between the pronotum and elytra in dorsal and lateral view. The interspersed, white elytral setae of these three species exhibit varying degrees of apical expansion, and can appear moderately to greatly explanate or spatulate in at least some, but not all, specimens.

Minyomerus franko [JF2018] has been documented on spear globemallow Sphaeralcea hastulata [non-focal]. Minyomerus trisetosus [JF2015] is associated with broomweed Xanthocephalum [non-focal], creosote bush Larrea tridentata [non-focal] and snakeweed Gutierrezia [non-focal]. Minyomerus caseyi has no known plant associations. It is therefore possible that the divergence of M. franko [2018] was facilitated by differences in host plant preference. However, this remains unlikely given the generalist feeding habits of congenerics.

Alternatively, the speciation sequence in the M. caseyi–M. franko clade [JF2018] may correspond to vicariance events. Minyomerus trisetosus [JF2015] inhabits a broad swath of the northern Chihuahuan Desert, whereas M. franko [JF2018] and M. caseyi [JF2015] are exclusively encountered in the southern Chihuahuan Desert. MaxEnt predicts overlapping species distributions for the latter two species. However, the documented localities of these two species pertain to distinct biogeographic regions. Minyomerus franko [JF2018] has only been collected in the valleys of the Sierra Madre Oriental range, whereas M. caseyi [JF2015] is found along the western edge of this range, in the eastern portion of the Central Mexican Plateau. Additional occurrence records are needed to clarify the spatial extents of these species’ distributions, and thus draw more robust inferences regarding their endemicity.

Conclusions

Through addition of four herein described species, the entimine [non-focal] genus Minyomerus [JF2018] is expanded to include 21 species. We predict that additional undescribed species of Minyomerus [JF2018] exist throughout the North American deserts, given the narrow endemicity patterns of many members of the genus. Furthermore, we believe that sampling in poorly-sampled locales, particularly in the northwestern United States and in northern Mexico, will yield new evolutionary insights for this group. New molecular data can strengthen phylogenetic hypotheses and provide estimates regarding the timing of diversification of Minyomerus [JF2018], thereby testing our current inference of an origin in central Mexico. Another research direction should focus on the reproductive behavior of certain species suspected to be parthenogenetic; including rearing and karyotyping. Finally, the validity of the genus Minyomerus [JF2018] as a member of the Tanymecini [non-focal], and its relationships to other Entiminae [non-focal], remain uncertain.

Supplemental Information

Supplemental Information 1 Raw matrix data, bootstrap replicates, bremer support values, and most-parsimonious tree.

Click here for additional data file.

Supplemental Information 2 Explanation of RCC-5 alignment approach.

Click here for additional data file.

Supplemental Information 3 Minyomerus INT 2018/2015 Class (txt).

Input constraints for the Minyomerus [JF2018]/[JF2015] rank-only classification alignment.

Click here for additional data file.

Supplemental Information 4 Minyomerus INT 2018/2015 Class (PDF).

Input visualization for the SI2A input.

Click here for additional data file.

Supplemental Information 5 Minyomerus INT 2018/2015 Class mir.

Set of 114 Maximally Informative Relations (MIR) for the SI2A input.

Click here for additional data file.

Supplemental Information 6 Minyomerus INT 2018/2015 Class 0 mnpw.

Alignment visualization for the SI2A input.

Click here for additional data file.

Supplemental Information 7 Minyomerus INT 2018/2015 Phylogeny.

Input constraints for the Minyomerus [JF2018]/[JF2015] phylogeny alignment – whole-concept resolution with overlap.

Click here for additional data file.

Supplemental Information 8 Minyomerus INT 2018/2015 Phylogeny.

Input visualization for the SI3A input.

Click here for additional data file.

Supplemental Information 9 Minyomerus INT 2018/2015 Phylogeny mir.

Set of 925 Maximally Informative Relations (MIR) for the SI3A input.

Click here for additional data file.

Supplemental Information 10 Minyomerus INT 2018/2015 Phylogeny 0 mnpw.

Alignment visualization for the SI3A input.

Click here for additional data file.

Supplemental Information 11 Minyomerus INT 2018/2015 Phylogeny.

Input constraints for the Minyomerus [JF2018]/[JF2015] phylogeny alignment -- split-concept resolution.

Click here for additional data file.

Supplemental Information 12 Minyomerus INT 2018/2015 Phylogeny.

Input visualization for the SI4A input.

Click here for additional data file.

Supplemental Information 13 Minyomerus INT 2018/2015 Phylogeny mir.

Set of 925 Maximally Informative Relations (MIR) for the SI4A input.

Click here for additional data file.

Supplemental Information 14 Minyomerus INT 2018/2015 Phylogeny 0 mncb.

Alignment visualization for the SI4A input.

Click here for additional data file.

The authors are grateful to Robert Anderson (CMNC), Ed Riley and John Oswald (TAMU), and Lourdes Chamorro (USNM) for their assistance and provision of specimens used in this study. The authors also thank Salvatore Anzaldo, Andrew Johnston, Sangmi Lee and other ASUHIC members for their assistance in procuring, treating, and maintaining specimen loans upon entry into the ASUHIC.

Additional Information and Declarations

Competing Interests

Author Contributions

Data Availability

New Species Registration

The authors declare that they have no competing interests.

M. Andrew Jansen conceived and designed the experiments, performed the experiments, analyzed the data, contributed reagents/materials/analysis tools, prepared figures and/or tables, authored or reviewed drafts of the paper, approved the final draft.

Nico M. Franz conceived and designed the experiments, performed the experiments, analyzed the data, contributed reagents/materials/analysis tools, prepared figures and/or tables, authored or reviewed drafts of the paper, approved the final draft.

The following information was supplied regarding data availability:

Morphological matrix and RCC-5 alignment data including results of prior analyses from the 2015 revision of this genus are available at Data from: Descriptions of four new species of Minyomerus Horn, 1876 sec. Jansen & Franz, 2018 (Coleoptera: Curculionidae), with notes on their distribution and phylogeny, http://doi.org/10.5061/dryad.j8c5j18.

The following information was supplied regarding the registration of a newly described species:

Publication LSID: urn:lsid:zoobank.org:pub:0AEE5733-06D1-401F-88C9-0D5232FBFC7A

Minyomerus ampullaceus: urn:lsid:zoobank.org:act:24943E17-F20E-4E3C-A3A1-A1D4D907B48E

Minyomerus franko: urn:lsid:zoobank.org:act:F8C0153E-DF0E-40E0-AF31-EBEA7075D06D

Minyomerus sculptilis: urn:lsid:zoobank.org:act:EA0B1AD9-68F2-4409-A0F8-903B0DA0FFF9

Minyomerus tylotos: urn:lsid:zoobank.org:act:10CD3562-5969-4BCF-ACFE-BB0E5E2BF9A6

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
