# Peer review of "Descriptions of four new species of Minyomerus Horn, 1876 sec. Jansen & Franz, 2018 (Coleoptera: Curculionidae), with notes on their distribution and phylogeny"

_PeerJ, doi:10.7717/peerj.5633_

## Round 0.1 · original submission · Minor Revisions

Dear Michael and Nico,

I have received three reviews of your paper. Please, address them and send me back the corrected version asap.

Best regards

Juan Morrone

·

Basic reporting

The manuscript is written in clear, unambiguous and professional English language. The main objective of this work is to describe four new species of the weevil genus Minyomerus (Curculionidae, Entiminae, Tanymecini), previously revised by the authors of the article, and to integrate the new findings in an updated key and a phylogenetic analysis based on morphological characters.
The article is well organized and the structure conforms to PeerJ Standards. The introduction includes the necessary information and background on the taxon under study. The literature is well referenced and relevant.
The taxonomic study was based on a sufficient number of specimens collected by the authors and borrowed from different entomological collections. The descriptions are very detailed and the structures of diagnostic value are illustrated with very good photographs (dorsal and lateral habitus, mouthparts, sternites VIII of female, spermathecae). All figures are relevant, high quality and well labeled. The literature is well referenced and relevant.

Experimental design

The new species described have been registered in ZooBank, and the information on LSID’s in provided.
A cladistic analysis was performed on a data matrix of 52 morphological characters and 26 terminal taxa (5 outgroups and 21 ingroups), to see the position of the new species in the context of Minyomerus and related genera. The selection of characters and character states was correct. The most parsimonious tree was found using TNT, and NONA was used for the optimization of the characters. The tree with the characters optimized was conveniently illustrated.
The cladistics methods are sufficiently detailed and the information given can be replicated.
The potential distribution of each new species was assessed applying an ecological modeling approach, using MAXENT, and was shown in maps.
The authors adopted the taxonomic concept approach, including the use of taxonomic concept labels and Region Connection Calculus (RCC–5) articulations and alignments. I am not very familiar with this approach and I find it a bit confusing for reading the results, but I understand that the authors are leaders in this matter, not yet commonly used in the taxonomic papers. The details of the RCC–5 alignment are given in free text form in the Supplemental Information, which also describes the content of the data input and output files.

Validity of the findings

The work provides new taxonomic information. The descriptions of the new species are well justified, same as the results of the cladistics analysis. In the discussion, the authors compared the results of the new phylogeny with those previously obtained. They discussed on the intrageneric relationships and the phylogenetic position of each species in a historical-biogeographic context. In find it interesting and not excessively speculative, however, I have some difficulties in following the indication relative to the taxonomic concept approach.
Conclusions are well stated, linked to original research question and limited to
supporting results

Additional comments

The paper provides new basic taxonomic information, consisting on a description of four new weevil species, rigorously described and illustrated. The phylogenetic placement of these species is analyzed and their geographic ranges are discussed in relation to those of their putative sister taxa, based on the results of a niche modeling analysis. The methods applied are classical and correct. All the figures are appropriate and necessary.
In my opinion, the taxonomic concept approach applied in this paper is somewhat confusing, as well as the supplementary information relative to this subject, but I am not an expert on this matter.

·

Basic reporting

This is a comprehensive, well-written followup to the authors previous paper revising the genus Minyomerus. I have made a few minor comments in the text of the manuscript PDF. I am quite sure that the single specimen of M. ampullaceus is from the stomach/gut contents of a roadrunner. The USNM was quite active in the mid 1900's in supporting identification of gut contents from various species of birds. I have a new Phyxelis from Louisiana known only from USNM specimens from quail stomach contents and the weevil Glaphyrometopus ornithodorus was described from a series of USNM specimens from meadowlark stomach contents. As such, the paragraph related to the natural history of this species should be rewritten.
The phylogenetic work appears to be well supported and discussed with all characters and states well-explained. As the authors know, I am not a fan of the taxonomic concept approach but I can understand the authors rationale for its use. The authors have done a nice job of integrating the 4 new species into the key and phylogeny and presenting these results.

Experimental design

See above.

Validity of the findings

See above.

Additional comments

See above.

·

Basic reporting

No comment.

Experimental design

I suggested in the pdf itself that some information may need to be added in the methods section regarding additive/non-additive coding of multistate characters.

Validity of the findings

No comment.

Additional comments

This is a thorough study that aims to describe and phylogenetically place 4 new species of Minyomerus. The use of occurrence data to predict distribution ranges for each species is a much welcome approach to this type of study.

---

## Round 0.2 · accepted · Accept

Dear Michael and Nico: The revision is OK so I am suggesting that your paper is accepted. Congratulations!!!!

#